# Antarctic atmospheric Richardson number from radiosoundings measurements and AMPS

Qike Yang[1,2,3], Xiaoqing Wu[1,2,3], Xiaodan Hu[1,2,3], Zhiyuan Wang[1,3], Chun Qing[1,2,3], Tao Luo[1,2,3], Pengfei Wu[1,2,3], Xianmei Qian[1,2,3], and Yiming Guo[1,2,3]

[1]Key Laboratory of Atmospheric Optics, Anhui Institute of Optics and Fine Mechanics, HFIPS, Chinese Academy of Sciences, Hefei 230031, China
[2]Science Island Branch of Graduate School, University of Science and Technology of China, Hefei 230026, China
[3]Advanced Laser Technology Laboratory of Anhui Province, Hefei 230037, China

**Correspondence:** Xiaoqing Wu (xqwu@aiofm.ac.cn)

**Abstract.** Monitoring a wide range of atmospheric turbulence over the Antarctic continent is still tricky, while the atmospheric Richardson number ($Ri$; a valuable parameter which determines the possibility that turbulence could be triggered) is easier to obtain. The Antarctic atmospheric $Ri$, calculated from the potential temperature and wind speed, was investigated using the daily results from the radiosoundings and forecasts of the Antarctic Mesoscale Prediction System (AMPS). Radiosoundings
for a year at three sites (McMurdo, South Pole, and Dome C) were used to quantify the reliability of the AMPS forecasts. The AMPS-forecasted $Ri$ can identify the main spatiotemporal characteristics of atmospheric turbulence over the Antarctic region. The correlation coefficients ($R_{xy}$) of $\log_{10}(Ri)$ at McMurdo, the South Pole, and Dome C are 0.71, 0.59, and 0.53, respectively. The $Ri$ was generally underestimated by the AMPS and the AMPS could better capture the trend of $\log_{10}(Ri)$ at relatively unstable atmospheric conditions. The seasonal median of $\log_{10}(Ri)$ along two vertical cross-sections of the AMPS
forecasts are presented, and it shows some zones where atmospheric turbulence can be highly triggered in Antarctica. The $\log_{10}(Ri)$ distributions appear to be reasonably correlated to some large-scale phenomena or local-scale dynamics (katabatic winds, polar vortices, convection, gravity wave, etc) over the Antarctic plateau and surrounding ocean. Finally, the $\log_{10}(Ri)$ at the planetary boundary layer height ($PBLH$) were calculated and their median value is 0.316, this median value, in turn, was used to estimate $PBLH$ and agree well with the AMPS-forecasted $PBLH$ ($R_{xy} > 0.69$). Overall, our results suggest that
the estimated $\log_{10}(Ri)$ by AMPS are reasonable and the turbulence conditions in Antarctica are well revealed.

## 1 Introduction

The Richardson number ($Ri$) is a valuable parameter for giving insight into atmospheric stability; it combines both thermo-dynamic and dynamic profiles, which provides us with valuable insights into turbulent heat fluxes (Town and Walden, 2009) and the probability that optical turbulence (Yang et al., 2021, 2022) can be triggered in Antarctica. However, the measure-
ments of atmospheric properties in Antarctica are sparse compared to those in the mid-latitudes and tropics. Atmospheric models have been developed to overcome this limitation (Meso-NH by Lascaux et al., 2009; Polar WRF by Bromwich et al., 2013; MAR by Gallée et al., 2015), allowing researchers to investigate atmospheric variability beyond observational

coverage, even for forecasting atmospheric parameters in the future. The Antarctic Mesoscale Prediction System (AMPS; https://www2.mmm.ucar.edu/rt/amps/) runs a real-time atmospheric model and provides numerical forecasts for Antarctica. The performances of AMPS in forecasting temperature, wind, precipitable water vapor, cloud, radiation, and heat flux have been examined in previous studies (Monaghan et al., 2005; Seefeldt et al., 2011; Vázquez B and Grejner-Brzezinska, 2012; Wille et al., 2016; Listowski and Lachlan-Cope, 2017; Hines et al., 2019). To our knowledge, using the AMPS to forecast $Ri$ has not been formally validated. Thus, this study will investigate the reliability of the estimated $Ri$ using AMPS forecasts. The atmospheric model employed for AMPS is the Polar version of the Weather Research and Forecasting (Polar WRF) model (Powers et al., 2012). Polar WRF (http://polarmet.osu.edu/PWRF/) has been modified for use in polar regions, for example, improving the representation of heat transfer through snow and ice (Hines and Bromwich, 2008; Hines et al., 2015). The Polar WRF has been used to simulate the $Ri$ at Dome A in Antarctica, and the simulated $Ri$ basically behaved as expected as the $Ri$ is generally large when the atmosphere is less turbulent (corresponding to the measured astronomical seeing is small; Yang et al., 2021) and performed well in estimating boundary layer height when compared with other methods (Yang et al., 2022).

Presently, monitoring a wide range of atmospheric turbulence over the Antarctic continent is tremendously difficult, but atmospheric $Ri$ is easier to obtain, as it can be calculated from the routine meteorological parameters (potential temperature and wind speed). However, few studies have evaluated atmospheric models to forecast $Ri$ in Antarctica, because of limited meteorological experiments here. Nevertheless, Geissler and Masciadri (2006) and Hagelin et al. (2008) used the European Centre for Medium-Range Weather Forecasts (ECMWF) analyses to calculate atmospheric $Ri$ in Antarctica. The ECMWF analyses were generated from the data assimilation using observations (P. Lönnberg, 1992), and can provide initial states for numerical models (such as Polar WRF). However, their research has some specific shortcomings (or problems that need further study): (1) They did not compare $Ri$ estimations from forecasts and measurements, while the forecast function is of great significance for practical application (e.g., astronomical observations, aviation safety, optical communication, etc). (2) How model errors of $Ri$ depend on atmospheric conditions has not been analyzed. (3) The correlations between turbulence conditions (indicated by $Ri$) and some large-scale phenomena or local-scale dynamics in Antarctica were not fully investigated. (4) A reference standard for judging the probability of triggering turbulence using the model-estimated $Ri$ was not given. To fill these gaps, the scientific goals of this paper are thus as follows:

1. To carry out a detailed comparison of potential temperature and wind speed (on which $Ri$ depends) in the atmospheric column, this study extends the model evaluations above two sites (Hagelin et al., 2008) to three sites (McMurdo, South Pole, and Dome C) over the Antarctic continent for an entire year. The three sites are considered representative, as the coast (McMurdo), flank (South Pole), and summit (Dome C) of the Antarctic continent will be compared using radiosoundings and AMPS forecasts.

2. The radiosonde can measure meteorological parameters, which can estimate $Ri$. Using the AMPS-forecasted meteorological parameters, one also can obtain the $Ri$. Then, a comparison of $Ri$ estimated from measurements and forecasts can be achieved, allowing us to evaluate the reliability of AMPS-forecasted $Ri$ in giving insight into the atmospheric turbulence in Antarctica. In addition, we investigated how the discrepancies between the models and measurements depend on the atmospheric conditions.

3. Two vertical cross-sections for $Ri$ will be given, which may provide a better perspective on the turbulence conditions in both vertical and horizontal dimensions, instead of only focusing on the vertical dimension (or atmospheric column; e.g., Geissler and Masciadri, 2006; Hagelin et al., 2008). This will help to identify regions and periods that are favorable for triggering atmospheric turbulence in Antarctica. Moreover, this will enable us to correlate the $Ri$ distribution with some large-scale phenomena or local-scale dynamics (katabatic winds, polar vortices, convection, gravity wave, etc.) in Antarctica, and the underlying physical processes of Antarctic atmospheric turbulence will be investigated.

4. The Planetary Boundary Layer Height ($PBLH$, within which the atmosphere is generally turbulent) can be estimated using a critical value of $Ri$, typically 0.25 (Holtslag et al., 1990; Pietroni et al., 2012; Petenko et al., 2019). However, this critical value depends on the vertical resolution of data (Troen and Mahrt, 1986; Holtslag et al., 1990), and may be different for the AMPS grid resolution. Then the $Ri$ at the AMPS-forecasted $PBLH$ ($Ri_{PBLH}$) was obtained as a reference standard for judging whether the atmosphere is likely to be laminar flow ($Ri>Ri_{PBLH}$) or turbulent flow ($Ri<Ri_{PBLH}$) when using the AMPS-forecasted $Ri$.

In Sect. 2, we present the experimental data and atmospheric model used in this study, with an explanation of their main characteristics. In Sect. 3, the Richardson number is introduced. In Sect. 4, we compare AMPS forecasts to radiosoundings and analyze the atmospheric turbulence conditions in Antarctica. Sect. 5 summarizes the main findings and primary takeaways of this study.

## 2  Data and model

### 2.1  Radiosoundings

Daily radiosounding measurements at McMurdo (MM) and the South Pole (SP) are available at the Antarctic Meteorological Research Center (AMRC; ftp://amrc.ssec.wisc.edu/pub). For Dome C (DC), one can obtain the measurements at the Antarctic Meteo-Climatological Observatory (http://www.climantartide.it). The altitudes of the three sites are 9 m (MM), 2839 m (SP), and 3239 m (DC), where the altitudes correspond to the heights of the radiosondes at the time of launch. Their locations are shown in Fig. 1. Dome A (DA) is also marked in Fig. 1, which is the highest location (4083 m) on the Antarctic plateau and the atmospheric conditions above it will also be analyzed in this study (Sect. 4.2.3). The radiosonde-measured meteorological parameters include pressure, temperature, wind speed, and wind direction; one year (from 2021 March to 2022 February) of these meteorological parameters was used in this study. Generally, the radiosonde was launched once a day at the same hour (sometimes twice a day at MM and SP). In total, 518, 508, and 340 profiles are available at MM, SP, and DC from 2021 March to 2022 February.

The radiosonde instrumentation used during this measurement period was the Vaisala RS41 (Technical data: https://www.vaisala.com/en/products/weather-environmental-sensors/upper-air-radiosondes-rs41). The accuracy and uncertainty of the radiosonde measurements are listed in Table 1. Vaisala RS41 radiosondes have gradually replaced an older version (Vaisala RS92) starting in late 2013. These two radiosondes agree well with global average temperature differences <0.1-0.2 K in the lower stratosphere, but RS41 appears to be less sensitive than RS92 to changes in solar elevation angle (Sun et al., 2019). Be-

**Table 1.** Main technical specifications of the radiosonde RS41.

| Measuring element | System resolution | System uncertainty | Data resolution* |
|---|---|---|---|
| Temperature | 0.01°C | 0.15°C (> 100 hPa) | 0.1°C |
| | | 0.30°C (<100 hPa) | |
| Pressure | 0.01 hPa | 0.5 hPa (> 100 hPa) | 0.1 hPa |
| | | 0.3 hPa (3-100 hPa) | |
| Wind speed | 0.1 m s$^{-1}$ | 0.15 m s$^{-1}$ | 0.1 m s$^{-1}$ (McMurdo) |
| | | | 0.1 kts (South Pole) |
| | | | 0.1 m s$^{-1}$ (Dome C) |
| Wind Direction | 0.1 deg | 2 deg | 0.1 deg (McMurdo) |
| | | | 1 deg (South Pole) |
| | | | 1 deg (Dome C) |

*Resolution in the files that are available for download from the Web (ftp://amrc.ssec.wisc.edu/pub, http://www.climantartide.it).

sides, RS41 (1-1.5% dry bias) has better performance than RS92 (3-4% dry bias) relating to the infrared atmospheric sounding interferometer as a practical reference (Sun et al., 2021). Near-global radiosonde measurements have been used to calculate the Richardson number and derive the boundary layer height, which is positively correlated with the results of four reanalysis products (Guo et al., 2021).

In Antarctica, the radiosondes measure the atmosphere between the ground and an altitude of 10–25 km (low in winter and high in summer) with a typical ascent rate of 5 m s$^{-1}$, and a logging frequency of 1 Hz; then the vertical resolution is approximately 5 m.

## 2.2  AMPS

The AMPS can forecast meteorological parameters in four-dimensional space-time in Antarctica, which can be used for com-
parison with the radiosonde measurements. The AMPS grid system consisted of a series of nested domains with 60 vertical levels. This study used grid 2 fields (d02; 8 km horizontal resolution) that covered the entire Antarctic continent (similar to Hines et al., 2019), as shown by the white square in Fig. 1. However, the contributions from the nested grid with higher horizontal resolution (d03: 2.67 km; d05: 0.89 km; d06: 2.67 km) are not entirely lost, as the AMPS used a two-way nested run and the nest (e.g., d03) feeds its calculation back to the coarser domain (e.g., d02). The original WRF output
files for each AMPS grid were saved in a rolling archive (one can find how to download the original WRF output files at https://www2.mmm.ucar.edu/rt/amps/information/amps_esg_data_info.html). This study used the AMPS outputs (in original WRF format) from the daily AMPS forecasts that began at 12:00 UTC. Parish and Waight (1987) showed large adjustments to the boundary layer fields above an ice sheet before the numerical model began to stabilize after about 10 h. Then, some studies (Hines and Bromwich, 2008; Hines et al., 2019) have discarded the first 12 h forecasts (so-called 12 h spin-up time). Thus, in

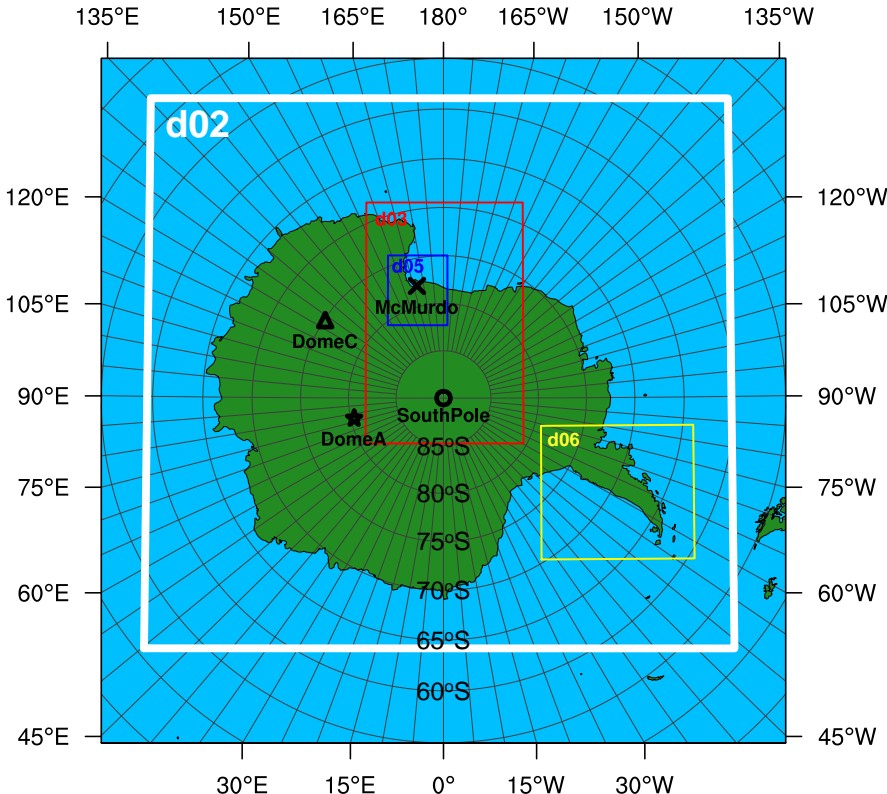

**Figure 1.** The five two-way interactive horizontal grids (d01, d02, d03, d05, and d06; information online at https://www2.mmm.ucar.edu/rt/amps/information/configuration/maps_2017101012/maps.html) used in the AMPS configuration. The locations of McMurdo (78°S, 167°E), South Pole (90°S, ... °E), Dome C (75°S, 123°E), and Dome A (80°S, 78°E) are shown by the cross, circle, triangle, and star, respectively. This study used grid 2 fields (d02; the white rectangle) that covered the entire Antarctic continent.

this study, only the 12-33-h forecasts from each of the AMPS simulations are combined into a year-long (2011 March to 2022 February) output field at 3-h intervals.

## 3   Theory of Richardson number

The Richardson number ($Ri$) is generally defined as (Richardson and Shaw, 1920; Chan, 2008):

$$Ri = \frac{g}{\theta} \frac{\partial\theta/\partial z}{[\partial u/\partial z]^2 + [\partial v/\partial z]^2} \tag{1}$$

Where $g$ is the gravitational acceleration (9.8 m s$^{-2}$), $\theta=T[1000/P]^{0.286}$ is the potential temperature (K), $T$ and $P$ are the temperature (K) and pressure (hPa) of air, respectively. As for wind shear term, $u$ and $v$ are the east-west and north-south

components of the wind (m s$^{-1}$). $z$ is the height (m) above the ground. To calculate $Ri$, a centered finite difference operation was used to estimate the gradient in Eq. (1).

The development of atmospheric turbulence was shown to be tightly correlated with the $Ri$. It can, therefore, be an essential
indicator of the turbulence characteristics in the atmosphere (Ma et al., 2020; Han et al., 2021; Yang et al., 2021). Atmospheric conditions are favorable for the occurrence of turbulence when $Ri$ is less than a critical value ($Ri_c$), and $Ri_c$ is typically chosen as 0.25. However, a larger $Ri_c$ should be used in a large-scale model (e.g., 0.5 has been employed by Troen and Mahrt, 1986).

In the results of this study, the logarithm of $Ri$, $\log_{10}(Ri)$, is presented instead of $Ri$ itself, because $Ri$ can vary by two or more orders of magnitude in the atmosphere.

## 4   Results and discussion

### 4.1   Potential temperature and wind speed

The AMPS forecasts are compared to radiosoundings from MM, SP, and DC to investigate the reliability of the AMPS forecasts over the Antarctic continent. The radiosoundings and AMPS forecasts used for this comparison were obtained from March 2021 to February 2022. To offer a more convincing result, data corresponding to the altitude at which radiosoundings reached
less than five times a season were discarded. In addition, the extracted AMPS forecasts used for comparison were from the nearest grid to the three sites, and the time difference between radiosoundings and AMPS forecasts larger than 1.5 hours was not used for comparison. Moreover, both radiosoundings and AMPS forecasts were linearly interpolated to the same height series (average annual altitude of the AMPS vertical grid; where the altitude of the AMPS grid may vary during the simulation as the AMPS uses the WRF hybrid vertical coordinate, information online at https://www2.mmm.ucar.edu/wrf/users/docs/
user_guide_v4/WRFUsersGuide.pdf) for each site. On the other hand, it should be noted that the near-surface radiosonde measurements could be less reliable, as it was just released from the operator's hand (or some machine). Hagelin et al. (2008) conclude that the radiosoundings are ∼1 K colder than the Automatic Weather Station at Dome C and ∼2 K at the South Pole. In this study, the radiosonde measurements in the first ∼10 m above the ground were not used. This is also because the first AMPS grid is ∼10 m above the ground.

The seasonal median difference of potential temperature (see the filled areas in Fig. 2) and wind speed (see the filled areas in Fig. 3) between radiosoundings and AMPS forecasts are presented. The missing value of median difference in the upper part of the atmosphere during JJA indicates that the radiosonde balloon does not reach as high an AGL (Above Ground Level) in winter as they do in summer, probably because the elastic material of the balloons is more fragile in cold seasons and easier to explode (Hagelin et al., 2008). The lack of measurements may be attributable to some large values of the median difference in
the top layer of the profile shown in Fig. 2 and 3, as the AMPS requires the assimilation data from measurements to initialize its numerical model, and the lack of measurements makes it more difficult for AMPS to simulate atmospheric changes that are close to reality.

Fig. 2 shows that the median difference for $\theta$ is of the order of 1 K in the first 5 km (except for the atmosphere layer in proximity to the ground). Above 5 km, the AMPS has obviously underestimated the $\theta$ at MM, while the forecasts at SP and DC

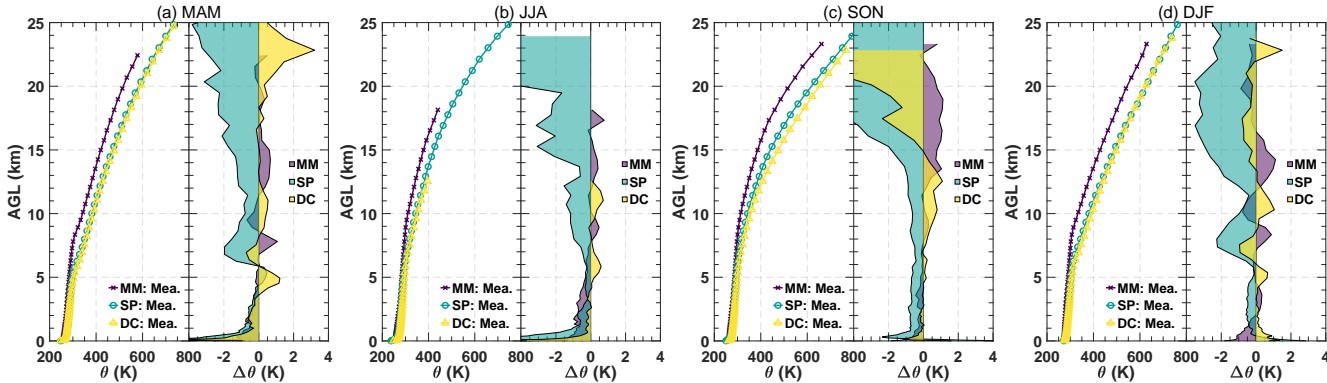

**Figure 2.** The seasonal median of potential temperature ($\theta$) estimated by the radiosonde measurements (solid lines) and potential temperature difference ($\Delta\theta$) calculated by the AMPS forecasts minus the radiosonde measurements, i.e. $\Delta\theta=\theta_{AMPS}-\theta_{Mea.}$ (filled areas). Fall: March-May (MAM); winter: June-August (JJA); spring: September-November (SON); summer: December-February (DJF).

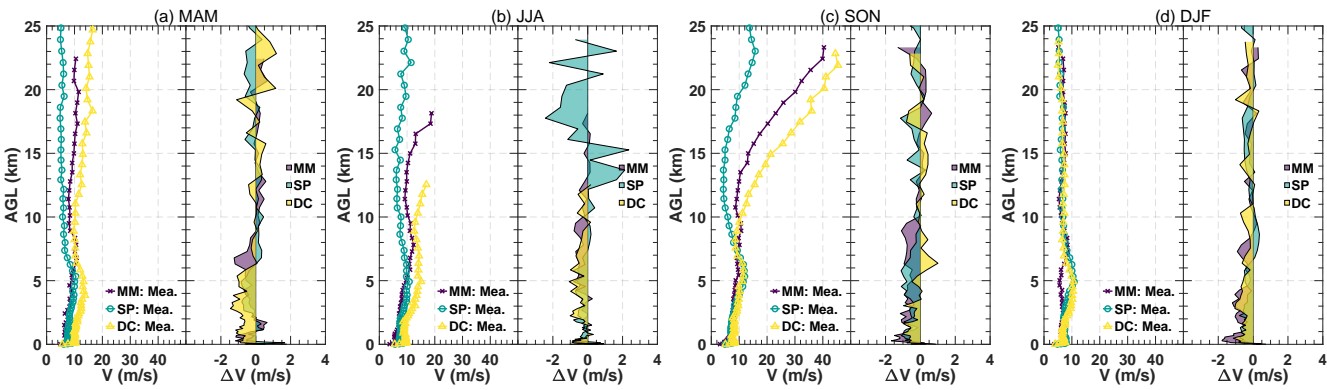

**Figure 3.** As in Fig. 2, but for wind speed $V$ (=$\sqrt{u^2+v^2}$), and $\Delta V=V_{AMPS}-V_{Mea.}$.

are closer to measurements. Fig. 3 shows that the measured wind speed profiles above 10 km at MM and DC are stronger during spring, indicating the occurrence of the Antarctic polar vortex (Boville et al., 1988). However, the change in wind speed above SP is not that obvious, because the Antarctic vortex is roughly pole-centered (Karpetchko et al., 2005). From the filled areas in Fig. 3, the AMPS forecasts appear consistent with the measurements, as the median difference in wind speed is generally $\sim$1 m s$^{-1}$ and has barely exceeded 2 m s$^{-1}$, whether the wind is strong or weak. In the first 10 km, most $\Delta V$ at the three sites

are less than 0, suggesting that the AMPS underestimated the wind speed. Table 2 shows the statistical evaluations of $\theta$ and $V$ forecasted by the AMPS. It seems the AMPS can well capture the trend of $\theta$ and $V$ as the correlation coefficient ($R_{xy}$) are all larger than 0.84.

## 4.2 Richardson number

### 4.2.1 Statistical analysis

To evaluate the performance of AMPS in forecasting the possibility of triggering turbulence over the Antarctic continent, the $Ri$ estimations between radiosoundings and AMPS forecasts will be compared. The calculated value of $Ri$ depends on the vertical resolution of meteorological parameters (Troen and Mahrt, 1986; Holtslag et al., 1990). Thus, the meteorological parameters from the radiosoundings and AMPS forecasts were interpolated into the same height series (as mentioned in Sect. 4.1) to calculate $Ri$. Where $\partial\theta/\partial z$ and $(\partial u/\partial z)^2 + (\partial v/\partial z)^2$ for calculating $Ri$ (see Eq. (1)) were both computed

using a centered finite difference operation, as we found that centered difference performed better than forward difference and backward difference (not shown), i.e., better consistency of $Ri$ between radiosoundings and AMPS forecasts can be achieved using centered difference.

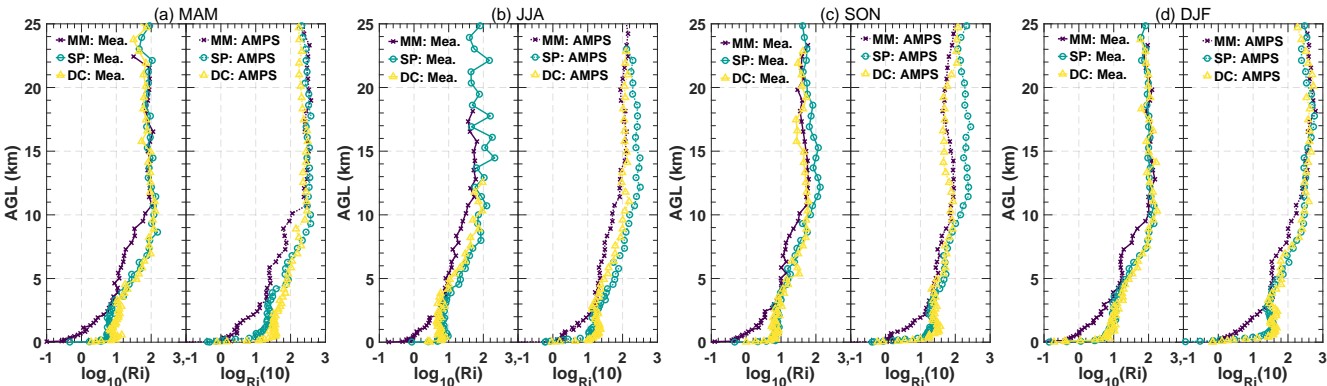

**Figure 4.** The seasonal median of $\log_{10}(Ri)$ estimated by the AMPS forecasts (solid lines) and the radiosonde measurements (dashed lines).

The seasonal median profiles of $\log_{10}(Ri)$ from radiosoundings and AMPS forecasts are shown in Fig. 4. However, the median differences are not presented like the $\theta$ and $V$. This is because, the $Ri$ value can vary massively (by two or more orders

of magnitude) in the atmosphere, and a precise quantification seems less plausible. Considering this, we initially intended to examine whether AMPS can reconstruct an accurate shape of $\log_{10}(Ri)$ profile (while median difference is not suitable for this purpose), and the results from radiosoundings and AMPS forecasts are both presented. Nevertheless, the model biases is by all means of great significance, and it will be discussed later (see Table 2 and Fig. 6). In Fig. 4, one can see that the AMPS-forecasted $Ri$ can identify that the atmosphere above MM tends to be more turbulent ($Ri$ is smaller) than SP and DC. In the

vertical height direction, the AMPS forecasts can roughly capture the height that can easily trigger turbulence. For example, one can observe that the $Ri$ from radiosoundings and AMPS forecasts both show small values very close to the ground at DC and the SP, which is per the fact that strong atmospheric turbulence is concentrated within the surface layer above the high plateau (Marks et al., 1999; Agabi et al., 2006). A very calm atmosphere ($Ri$ is large) at high altitudes is also consistent with

the results given by Travouillon et al. (2003), Aristidi et al. (2005), Trinquet et al. (2008), and Vernin et al. (2009). On the
other hand, the AMPS can well reconstruct the near-ground "convex-concave–convex" (hereafter "C-C-C") shaped $\log_{10}(Ri)$
profiles indicated by the radiosonde measurements (see more details in Fig. 5). In terms of time, the AMPS can forecast that
the free-atmosphere $Ri$ decreased during spring (SON), this decrease is obvious for MM and DC (where the wind speed are
significantly stronger during SON, as in Fig. 3).

**Table 2.** Statistical evaluations of the potential temperature ($\theta$), wind speed ($V$), and logarithmic Richardson number ($\log_{10}(Ri)$) forecasted
by the AMPS when compared with the results from radiosonde measurements.

| Season | McMurdo | | | | South Pole | | | | Dome C | | | |
|---|---|---|---|---|---|---|---|---|---|---|---|---|
| | MAM | JJA | SON | DJF | MAM | JJA | SON | DJF | MAM | JJA | SON | DJF |
| $\theta$: $R_{xy}$ | 0.99 | 0.99 | 0.99 | 0.99 | 0.99 | 0.99 | 0.99 | 0.99 | 0.99 | 0.99 | 0.99 | 0.99 |
| $\theta$: $Bias$ | -0.32 | -0.61 | -0.10 | -0.23 | -1.30 | -1.41 | -1.45 | -0.94 | -0.74 | -0.58 | -0.25 | 0.19 |
| $\theta$: $RMSE$ | 1.82 | 1.91 | 1.78 | 1.56 | 2.65 | 2.88 | 3.48 | 2.43 | 4.28 | 2.22 | 2.38 | 1.76 |
| $V$: $R_{xy}$ | 0.85 | 0.89 | 0.95 | 0.90 | 0.86 | 0.84 | 0.89 | 0.90 | 0.92 | 0.92 | 0.97 | 0.95 |
| $V$: $Bias$ | -0.25 | -0.25 | -0.59 | -0.67 | -0.16 | -0.29 | -0.64 | -0.40 | -0.62 | -0.43 | -0.24 | -0.50 |
| $V$: $RMSE$ | 3.16 | 3.63 | 3.23 | 2.81 | 2.52 | 2.90 | 2.69 | 2.37 | 2.50 | 2.94 | 2.69 | 1.89 |
| $\log_{10}(Ri)$: $R_{xy}$ | 0.75 | 0.65 | 0.68 | 0.77 | 0.61 | 0.50 | 0.56 | 0.70 | 0.51 | 0.45 | 0.50 | 0.66 |
| $\log_{10}(Ri)$: $Bias$ | 0.41 | 0.32 | 0.33 | 0.47 | 0.36 | 0.23 | 0.29 | 0.35 | 0.45 | 0.39 | 0.30 | 0.46 |
| $\log_{10}(Ri)$: $RMSE$ | 0.90 | 0.86 | 0.88 | 0.93 | 0.90 | 0.84 | 0.85 | 0.90 | 0.93 | 0.86 | 0.81 | 0.91 |

Quantitative analysis for the estimated $Ri$ from the numerical models was generally missed as it always varies dramatically
(e.g., Hagelin et al., 2008, who focused on the qualitative analysis). Nevertheless, quantitative analysis has been tried in this
study since that can give a precise evaluation of the forecast ability of AMPS. Then, the $R_{xy}$, mean bias ($Bias$; AMPS-
radiosonde), and root mean square error ($RMSE$) are calculated using the combined data of all profiles for each season. Where
the time difference between radiosoundings and AMPS forecasts was limited to less than 1.5 hours. Finally, the seasonal values
of the three statistical operators are calculated, as listed in Table 2. However, we want to emphasize that one should focus on the
value of $R_{xy}$ that reflects the tendency, instead of $Bias$ and $RMSE$, as a precise quantification remains in doubt (Hagelin et al.,
2008). The mean values of $R_{xy}$ for MM, SP, and DC over four seasons are 0.71, 0.59, and 0.53, respectively. The highest $R_{xy}$
is at MM for DJF (0.77) and the lowest is at DC for JJA (0.45). We found that these two cases correspond to the most unstable
and stable atmospheric conditions, their median $[\theta_{1000m}-\theta_{0m}]/[1000m-0m]$ equal to 0.0038 and 0.0721, respectively. This
suggests that the AMPS can better capture the trend of $\log_{10}(Ri)$ at a relatively unstable atmosphere. However, the $Bias$ is
the largest (0.47) in the most unstable case. This is because the AMPS overestimated the potential temperature gradient under
an unstable atmosphere (see Fig. 6a, which will be discussed later). For the stable atmosphere, the lowest $R_{xy}$ for $\log_{10}(Ri)$
seems to be consistent with the fact that model errors increase with increasing stability (Nigro et al., 2017).

Table 2 also shows an interesting result: the $R_{xy}$ of $\log_{10}(Ri)$ is higher when the $RMSE$ of $\theta$ and $V$ are smaller. Moreover, Hines et al. (2019) showed that using the Morrison microphysics scheme in the numerical model resulted in a smaller $RMSE$

for temperature and wind, than the default scheme (WSM5C) in AMPS. Therefore, we may conclude that replacing WSM5C with Morrison could improve the AMPS-forecasted $\log10(Ri)$. In other words, using Morrison may lead to higher $Rxy$ for $\log10(Ri)$, as it simulates dynamic stability with less variability (the $RMSE$ for temperature and wind could be smaller). On the other hand, larger $RMSE$ for $\theta$ and $V$ are mainly found during cold months (JJA, SON), indicating that winter dynamic stability is more variable (similar to Bromwich et al., 2013).

Table 2 summarises that the $\log_{10}(Ri)$ was overestimated by the AMPS at each site for every season (all $Bias$ are positive). This may be due to some local-scale dynamics not being represented properly (see Fig. 6, which will be discussed later). From another perspective, the model results were generally smoother than the measurements, and the atmosphere is less favorable for the occurrence of turbulence under slowly changing meteorological parameters, then the AMPS-forecasted $Ri$ could be larger.

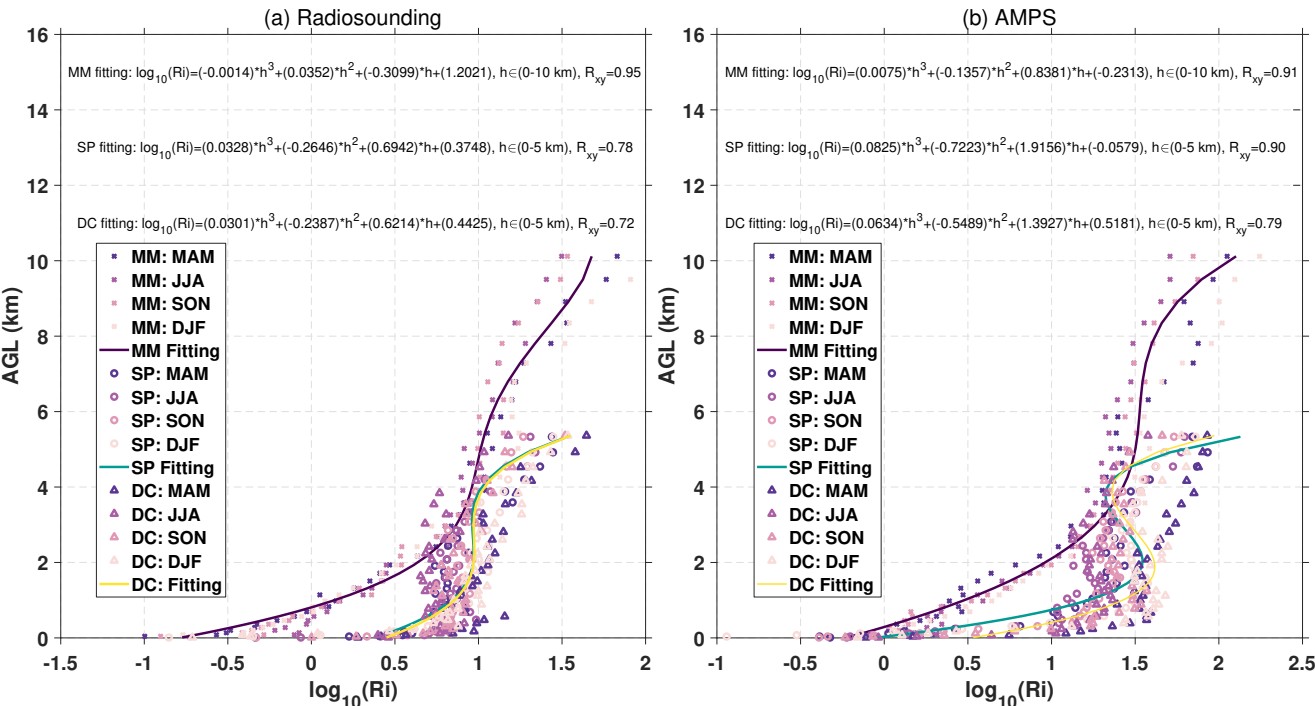

**Figure 5.** The polynomial curve fitting of near-ground median profiles of $\log_{10}(Ri)$ from Fig. 4. (a) the $\log_{10}(Ri)$ was estimated by the radiosonde measurements, (b) the $\log_{10}(Ri)$ was estimated by the AMPS forecasts.

The near-ground atmosphere in Antarctica is an important turbulence source, then an analytical function for $\log_{10}(Ri)$ profiles near the ground was fitted to better contextualize the results (as shown in Fig. 5). Fig. 5 shows that the near-ground $\log_{10}(Ri)$ profiles are the "C-C-C" shape. The "concave" structure in the "C-C-C" shape could be attributed by the near-ground

jet stream (Mihalikova et al., 2012). A cubic polynomial function was used (see the upper part of the plots in Fig. 5) instead of a logarithmic function, because the "C-C-C" shape seems hard for logarithmic function fitting. Moreover, each fitted curve

used all four-seasons data points in Fig. 4, as the seasonal variation are not too significant. Nevertheless, one can see more details about the temporal variation of $\log_{10}(Ri)$ near the ground in Sect. 4.2.3.

Fig. 6 shows the AMPS performance under different potential temperature gradient ($G = \partial\theta/\partial z$) and wind shear ($S = [(\partial u/\partial z)^2 + (\partial v/\partial z)^2]^{1/2}$). The statistical results presented in Fig. 6 were counted based on all the collected data points at the three sites (MM, SP, and DC) for an entire year. One can see that the $Ri$ was overestimated by the AMPS at unstable

atmosphere (see light blue bin in Fig. 6a), where the AMPS has overestimated the potential temperature gradient (i.e. $\Delta G > 0$). But for strong temperature inversion (see dark red bin in Fig. 6a), the AMPS has underestimated the $G$ and $Ri$. As for strong wind shear conditions (see dark red bin in Fig. 6b), when the $Ri$ is small (basically corresponding to a near-surface layer with a high probability of triggering strong turbulence, as in Fig. 4), the AMPS has underestimated the intensity of wind shear ($\Delta S < 0$). This may be caused by the AMPS has underestimated the wind speed near the ground (as in Fig. 3). In sum, if the

model aims for a more accurate forecast of $Ri$, the biases under these atmospheric conditions need to be corrected.

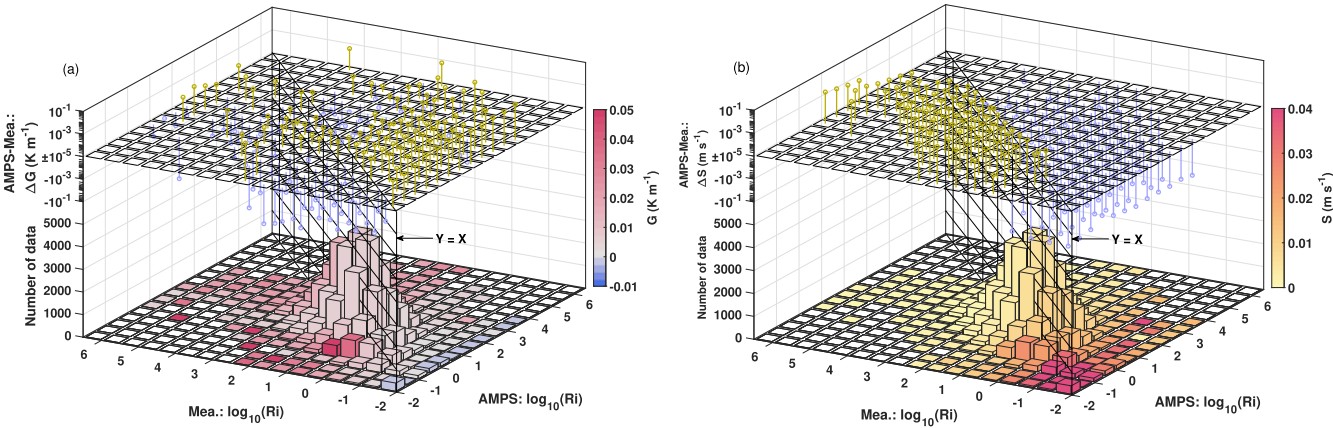

**Figure 6.** Performance of the AMPS under different atmospheric conditions. (a) and (b) are respectively the case of potential temperature gradient ($G = \partial\theta/\partial z$) and wind shear ($S = [(\partial u/\partial z)^2 + (\partial v/\partial z)^2]^{1/2}$). $G$ (color of the bin in (a)), $S$ (color of the bin in (b)), $\Delta G$ (stem above the bin in (a)) and $\Delta S$ (stem above the bin in (b)) are presented using the median value for each 0.5×0.5 bin of $\log_{10}(Ri)$.

### 4.2.2 Vertical cross-section

The results given in Sect. 4.2.1 show the AMPS can forecast the main tendency of $\log_{10}(Ri)$. Then, we consider that it is worth a try to use the AMPS-forecasted $\log_{10}(Ri)$ to comprehend the characteristics of atmospheric turbulence in Antarctica. The results of the AMPS-forecasted $\log_{10}(Ri)$ were presented through interpolation of the AMPS grid 2 field at two vertical

cross-sections, which provides us with a broader perspective on the probability of turbulence triggered in four-dimensional

space-time. One vertical cross-section is interpolated through the SP and DC, and another is through DA and MM, as shown in Fig. 7. The corresponding AMPS forecasts are shown in Figs. 8 and 9, respectively.

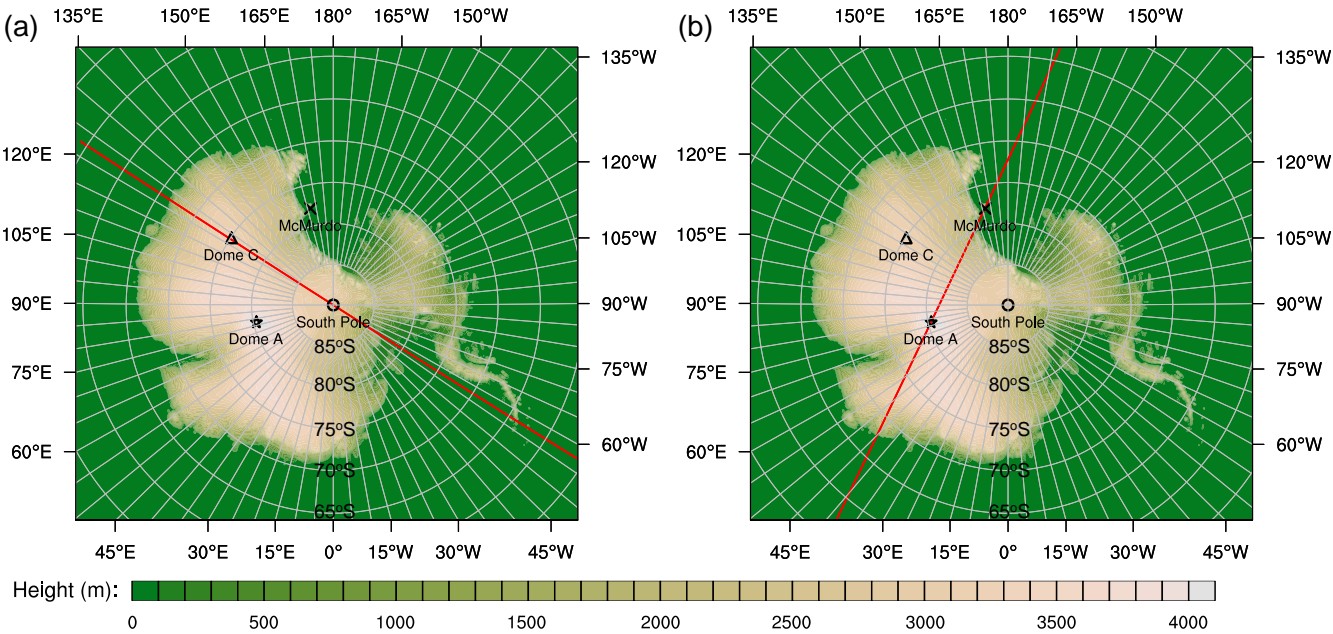

**Figure 7.** Two lines (marked by red lines) are used to create vertical cross-sections. (a) a line through DC and the SP, (b) a line through DA and MM. The color scale indicates the terrain height (m), where terrain fields are generated from the RAMP2 data set (https://nsidc.org/data/nsidc-0082/versions/2).

Figs. 8 and 9 show the seasonal median of the AMPS forecasts. The temperature and wind speed were lower above the Antarctic Plateau than over the ocean. In the polar winter (JJA), the temperature contours are dense near the ground above the interior Plateau, representing a strong surface-layer temperature inversion (such inversion has been observed by Yagüe et al., 2001; Argentini et al., 2013; Hu et al., 2019). The surface-layer wind speeds increase from the summit to the escarpment region (caused by the well-known katabatic wind over the surface slope area in Antarctica) and then decrease toward the coast, which is consistent with previous measurements (Ma et al., 2010; Rinke et al., 2012). The $Ri$ is obviously larger above the summits (e.g. DA and DC), suggesting the $PBLH$ could be thin, this agrees with the results from Swain and Gallée (2006); Bonner et al. (2010); Aristidi et al. (2015).

The results of $Ri$ distribution from the AMPS outputs provided us with valuable insights into the atmospheric turbulence in the Antarctic region (while using the radiosonde measurements is hard to do so). Here, we attempt to relate the features of atmospheric turbulence to some large-scale phenomena or local-scale dynamics over the Antarctic plateau and the ocean surrounding it: the shear-induced turbulence (katabatic winds, polar vortices), convection (cloud cooling, boundary layer convection), temperature inversion, and the wave-induced turbulence (orographic gravity waves, trapped lee waves, inertia-gravity

waves). Table 3 lists their possible functional areas that are marked in Figs. 8 and 9. This is dedicated to qualitatively evaluating the AMPS outputs and investigating the underlying physical processes of triggering atmospheric turbulence.

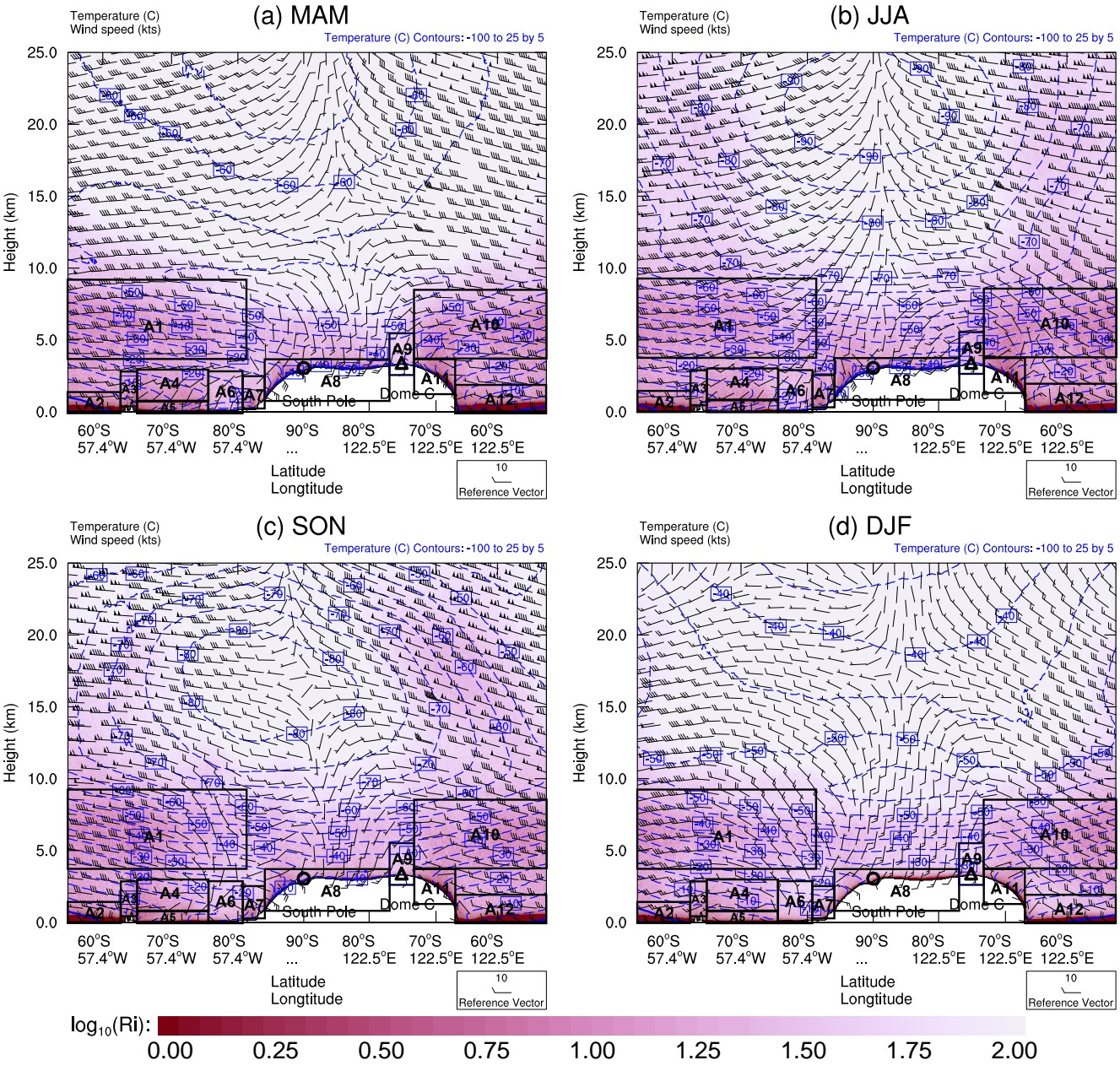

**Figure 8.** The seasonal median of temperature (instead of $\theta$), wind speed, and $\log_{10}(Ri)$ along the vertical cross-section through the South Pole (black circle) and Dome C (black triangle), as shown by the red line in Fig. 7a. The height (km) on the y-axis represents the elevation above sea level. The A$X$ in each plot are used to mark the possible functional areas of some atmospheric activities (as listed in Table 3).

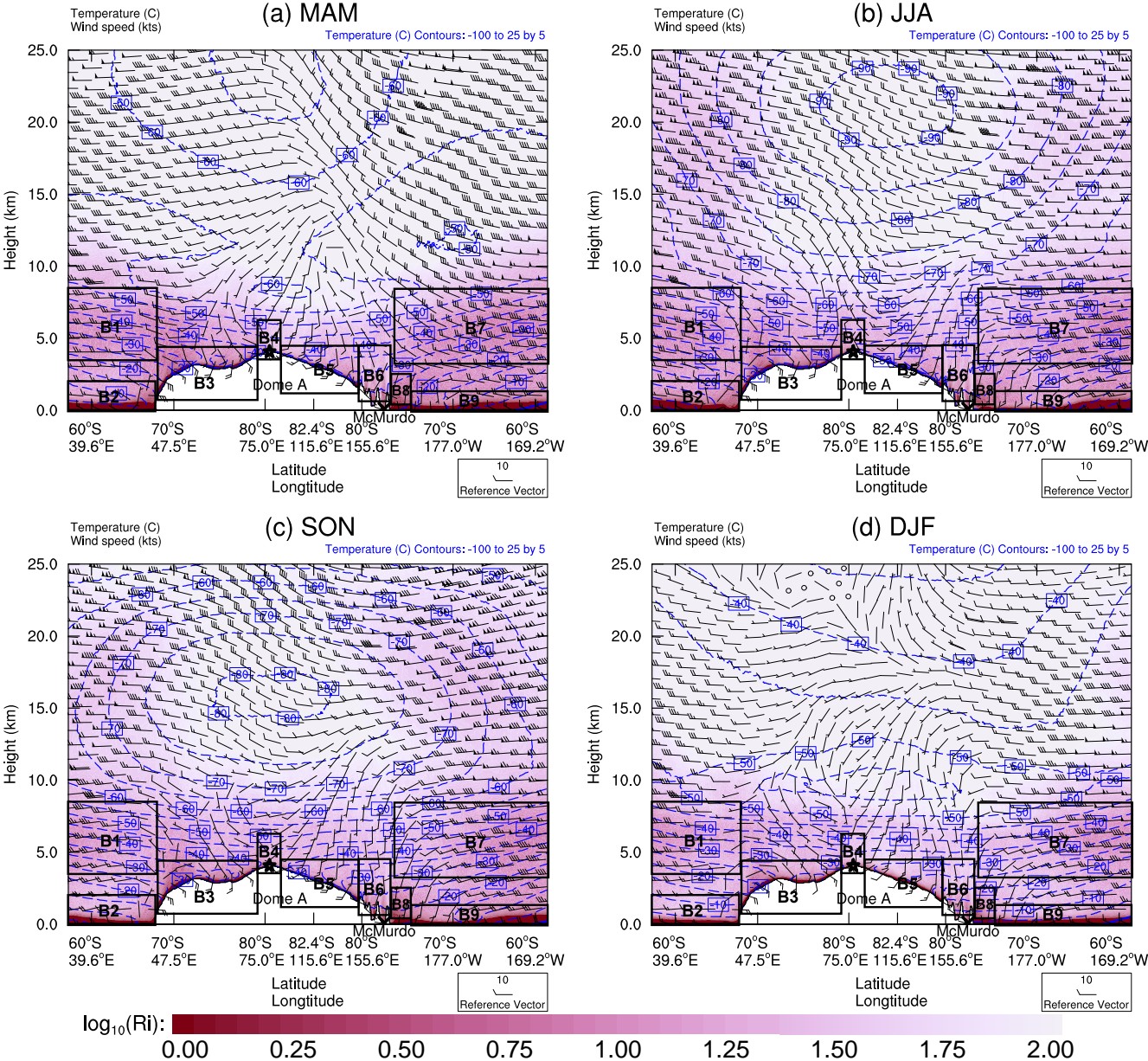

**Figure 9.** As in Fig. 8, but for the vertical cross-section through Dome A (black star) and McMurdo (black cross), as shown by the red line in Fig. 7b.

As a result of katabatic winds (Rinke et al., 2012), the near-surface wind speeds increase from the interior plateau to the steep slope (Figs. 8 and 9), which is driven by gravity. Strong winds can lead to strong wind shear and increased levels of

mechanical turbulence (e.g., Huang et al., 2021; Solanki et al., 2022), as one can see the surface layer with small $Ri$ at the

escarpment region (see A8 and A11 areas in Fig. 8, B3 and B5 areas in Fig. 9). Where the regions between the SP and DC (in A8 area) are also located on the slope (see Fig. 7a) and show a relatively small $Ri$ near the ground.

**Table 3.** The possible functional areas of some typical large-scale phenomena or local-scale dynamics over the Antarctic plateau and the ocean surrounding it.

| Atmospheric activities | Possible functional areas | | |
| --- | --- | --- | --- |
| | Marked areas in Fig. 8 | Marked areas in Fig. 9 | Contribution for triggering turbulence |
| Katabatic winds | A8, A11 | B3, B5 | Positive |
| Polar vortices | A1, A2, A5, A10, A12 | B1, B2, B7, B9 | Positive |
| Cloud cooling | A1, A10 | B1, B7 | Positive |
| Boundary layer convection | A2, A5, A12 | B2, B9 | Positive |
| Temperature inversion | A6, A9 | B4 | Negative |
| Orographic gravity waves | A3 | B6 | Positive |
| Trapped lee waves | A4, A7 | B8 | Positive |
| Inertia-gravity waves | A1, A10 | B1, B7 | Positive |

A strong polar vortex implies that the zonal winds are intense, and atmospheric turbulence is more prone to occur. The Antarctic polar vortex reaches its maximum intensity in the winter-spring season (Zuev and Savelieva, 2019a), which corre-
sponds to the relatively turbulent free atmosphere with low $Ri$ values over the ocean during JJA and SON (see Figs. 8 and 9). Moreover, the strongest zonal winds are located over the ocean (Zuev and Savelieva, 2019b), the interaction between the zonal wind and the ocean surface may generate wind shear and facilitate the development of turbulence (see areas A2 and A12 in Fig. 8, B2 and B9 in Fig. 9).

Cloud cooling refers to two kinds of cooling-induced turbulence in this study: Cloud Top Cooling (CTC) and Below Cloud-
base Turbulence (BCT). The CTC is contributed by radiative cooling, which could be one of the driving mechanisms of the mixed-layer turbulence (Deardorff, 1976). The BCT usually occurs below the bases of midlevel clouds accompanied by precipitation that does not reach the ground, cooling by evaporation or sublimation seems to contribute to the turbulence (Kudo, 2013; Kantha et al., 2019). In sum, regions with clouds may advance the development of turbulence. The cloud fraction observed by satellite lidar is higher above the ocean than the Antarctic plateau (Spinhirne et al., 2005; Saunders et al., 2009).
Thus, cloud may benfit small above the ocean (A1 and A10 areas in Fig. 8, plus the B1 and B7 areas in Fig. 9).

Boundary layer convection is generated by forcing from the ground, solar heating of the ground during sunny days causes thermals of warmer air to rise and convection will form (He et al., 2020), then the turbulence could be developed forced by buoyancy (Verma et al., 2017). The albedo of fresh snow over sea ice is very high, while that for open water is relatively small (Hines et al., 2015). Thus, solar heating will be much more stand out over open water and lead to the thermal convection boom.
This can be used to reasonably explain the results in Fig. 8, that the $Ri$ over ocean (A2 area) is smaller than over ice shelf (A6

area), and the A5 area can be regarded as a "transition region" (sea ice and open water could both exist) between them with an intermediate value of $Ri$.

The strength of the near-ground temperature inversion forecasted by the AMPS increases from the coast to the high interior, and its strength weakens during polar summer, such a phenomenon has also been observed in previous studies (Hudson and Brandt, 2005; Ma et al., 2010). The general increase in temperature-inversion strength was considered to correspond to a less turbulent atmosphere (when the boundary layer is shallower), owing to large stability suppressing turbulence. This corresponds to the larger $Ri$ in the summit area where a stronger temperature inversion occurred (see A9 area in Fig. 8 and B4 area in Fig. 9). There is a similar phenomenon occurred over the Ronne ice shelf (A6 area in Fig. 8), especially for JJA (when the temperature inversion is more obvious). Importantly, it should be noted that it is the range of turbulence (or $PBLH$) that would be suppressed by the temperature inversion and the turbulence intensity could be strong within the inversion layer (Petenko et al., 2019). For example, the turbulence above Dome C is mainly concentrated in the first tens of meters above the ground (Aristidi et al., 2015).

The development of Orographic Gravity Wave (OGW) is the interaction between near-surface wind and a mountain barrier (Lv et al., 2021; Zhang et al., 2022a, b), the OGW breaking could be a source of turbulence. Obviously, OGW can be triggered above the Antarctic Peninsula (A3 area in Fig. 8) and Transantarctic Mountains (B6 area in Fig. 9). But the atmosphere just above the top of the mountain seems to be laminar (e.g., see the larger value of $Ri$ in B6 area), this may be due to that the breaking of the OGW may not happen immediately after being generated above the mountains.

Trapped Lee Waves (TLW) belongs to OGW. Specially, TLW, as its name implies, tends to form on the lee side of mountains and turbulence may be developed in the downstream (Xue et al., 2022). Thus, the small $Ri$ in A4 area in Fig. 8 can be attributed by the TLW forced by the Antarctic Peninsula (see its position in Fig. 7a). It is the same case for B8 area in Fig. 9 (but forced by the Transantarctic Mountains). The katabatic winds could be linked to TLW and result in enhanced turbulence, This could explain the A7 area (Fig. 8) have small $Ri$ on the lee side of the mountain.

Inertia-Gravity Waves (IGW) are influenced by the Coriolis effect (increasing with wind speed), and the frequency of IGW is close to inertial frequency. IGW and Kelvin-Helmholtz instability (which can be characterized by the Richardson number) are generally presumed to be closely linked. At high latitudes, the IGW energy density's maxima occur at around 5 km AGL (Zhang et al., 2022b). This may suggest that the IGW can also be a contributor to the small $Ri$ above the ocean (A1 and A10 areas in Fig. 8, plus the B1 and B7 areas in Fig. 9).

In addition, one can see the temporal evolution of $Ri$ vertical cross-sections for a year from the video supplement (vertical cross-section through the red line shown in Fig. 7a: https://doi.org/10.5446/60761 and Fig. 7b: https://doi.org/10.5446/60760). It shows that the atmospheric conditions are variable, and a significant transition between laminar flow and turbulent flow could occur at any time. Some activities in Antarctica require a non-turbulent atmosphere, such as astronomical observations (Burton, 2010) and aviation safety (Gultepe and Feltz, 2019). Therefore, real-time forecasting of the Richardson number is important and helpful, rather than relying solely on the statistical results presented in this study. Furthermore, the video shows that atmospheric turbulence is likely to be triggered over the ocean, moving toward the Antarctic Plateau and weakening. This may be due to the obstruction of the high plateau, which creates a calm atmosphere above it.

### 4.2.3 Richardson number at the planetary boundary layer height

The Richardson number is used to determine the boundary layer height using a critical value (Troen and Mahrt, 1986; Holtslag et al., 1990; Pietroni et al., 2012). Thus, the critical value (or the value of $Ri$ at the $PBLH$, $Ri_{PBLH}$) is worth studying. In addition, previous studies have suggested that the $Ri_{PBLH}$ depends on the vertical resolution of the data (Troen and Mahrt, 1986; Holtslag et al., 1990). As for the resolution of the AMPS grid, it is necessary to recalculate $Ri_{PBLH}$ based on the AMPS outputs, since the value of $Ri_{PBLH}$ is a helpful reference for judging whether atmospheric turbulence is likely to be suppressed ($Ri>Ri_{PBLH}$) or developed ($Ri<Ri_{PBLH}$).

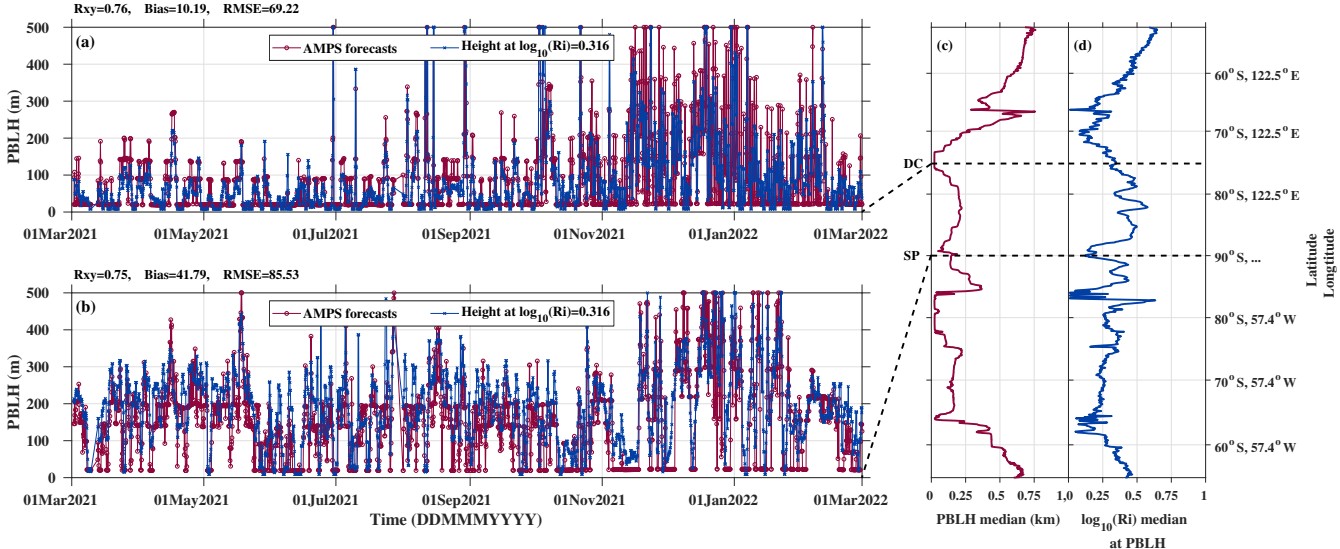

**Figure 10.** Temporal evolution of $PBLH$ directly forecasted by the AMPS (red circles) and estimated by the height corresponding to $\log_{10}(Ri)=0.316$ (blue crosses) at DC (a) and SP (b). Median annual $PBLH$ (c) and $\log_{10}(Ri)$ at the $PBLH$ (d) along the red line through DC and SP shown in Fig. 7a.

The planetary boundary layer scheme of Polar WRF in the AMPS was the Mellor-Yamada-Janjić (Janjić, 1994) scheme, and the ability of the AMPS to model the Antarctic boundary layer has been examined by (Wille et al., 2017). The MYJ scheme defines the $PBLH$ where turbulent kinetic energy decreases to a prescribed value of 0.1 m$^2$s$^{-2}$ (Xie et al., 2012). The AMPS forecasts include the values of $PBLH$. Figs. 10a and 11b show that the $PBLH$ directly forecasted by the AMPS was mostly less than 100 m in the summit (DC and DA) during the polar winter, such variation range is consistent with the SODAR observations (DC: Petenko et al., 2014; DA: Bonner et al., 2010); Fig. 10b also displays a result being in accordance with the SODAR observations at the SP, as the most $PBLH$ was shown to be within 100-300 m (Travouillon et al., 2003). Thus, the AMPS-forecasted $PBLH$ is considered to be realistic.

Figs. 10c and 11c show the median annual $PBLH$ forecasted by AMPS along the red lines in Figs. 7a and 7b, respectively. A thin $PBLH$ over the plateau can be observed, especially at the Domes (e.g., DA and DC), which is consistent with previous

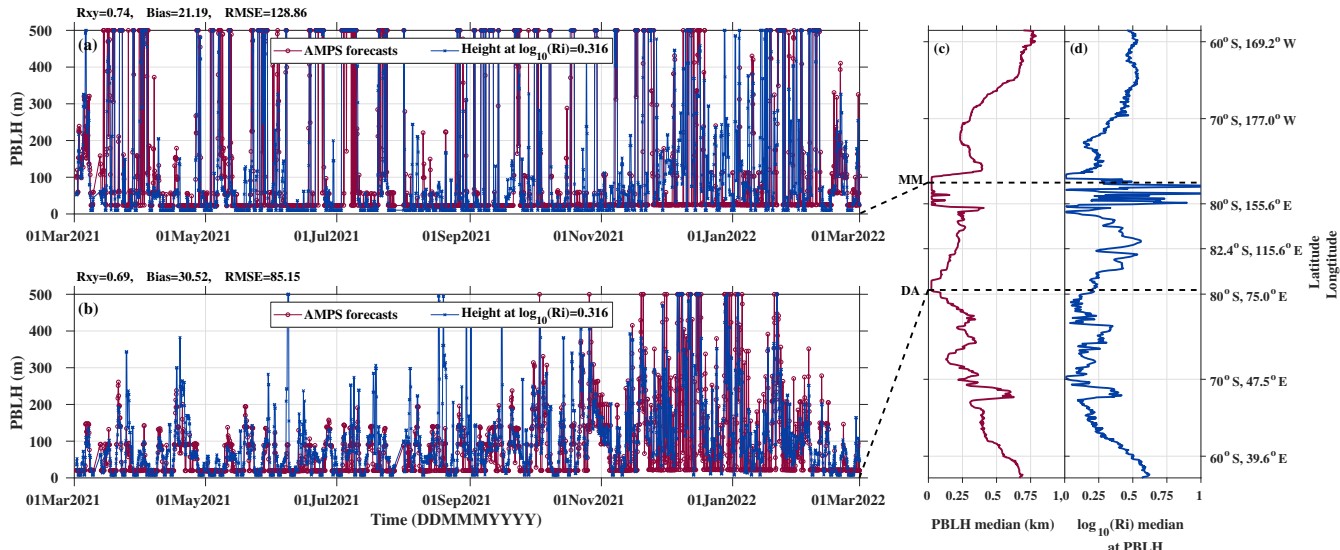

**Figure 11.** As in Fig. 10, but for data above MM and DA, plus the red line through MM and DA shown in Fig. 7b.

studies (Swain and Gallée, 2006). In contrast, a thick $PBLH$ is shown near the escarpment region (e.g., $\sim 68°$S, $122.5°$E in Fig. 10c; this corresponds to the relatively low $Ri$ near the ground in A11 area in Fig. 8).

The $\log_{10}(Ri_{PBLH})$ was computed using linear interpolation between the grid with height equals to the AMPS-forecasted $PBLH$. The median value of $\log_{10}(Ri_{PBLH})$ from the combined data of two vertical cross-sections (i.e. Figs. 10d and 11d) was calculated as 0.316, i.e., median $\log_{10}(Ri_{PBLH})$ equals to 0.316 (or $Ri_{PBLH}$=2.07). However, some researchers have employed $Ri_{PBLH}$=0.25 when the radiosonde measurements with a higher vertical resolution was used (e.g. at Dome C; Pietroni et al., 2012). Here, the larger $Ri_{PBLH}$ for the AMPS forecasts may be caused by the coarse vertical grid resolution (as

implied from Troen and Mahrt, 1986) and its data smoothness (as mentioned in Sect. 4.2.1; the AMPS forecasts show larger $Ri$ than the radiosoundings even though they have already been interpolated to the same vertical grid).

     To test the credibility of the critical value (i.e. $\log_{10}(Ri_{PBLH})$=0.316), $PBLH$ was also derived as the height where the AMPS-forecasted $\log_{10}(Ri)$ decreases to 0.316. The $R_{xy}$, $Bias$, and $RMSE$ of $PBLH$ between the estimations using the critical value (blue lines in Figs. 10a-b and 11a-b) and the direct forecasts of AMPS (red lines in Figs. 10a-b and 11a-b) are

depicted in the top left of the plot, where $Bias$ indicates the former minus the latter. It appears that the values of $R_{xy}$ (all larger than 0.69) are almost satisfactory, then we may conclude that $\log_{10}(Ri_{PBLH})$=0.316 is a reliable critical value for judging the behavior of atmospheric turbulence. The atmosphere layer could be considered turbulent for $\log_{10}(Ri_{PBLH})$<0.316 (when the turbulence intensity could be comparable to that within the boundary layer). However, this critical value may only be valid for using the AMPS forecasts.

## 5 Conclusions

We have examined the ability of AMPS to forecast the Richardson number in the Antarctic atmosphere. This includes evaluating the accuracy of meteorological parameters ($\theta$ and $V$, on which the $Ri$ depends), and comparing the $\log_{10}(Ri)$ estimations between radiosoundings and AMPS forecasts. In addition, the analysis of atmospheric $\log_{10}(Ri)$ over the entire Antarctic continent and the ocean surrounding it, was presented on an annual time scale. Finally, the $\log_{10}(Ri)$ at the Planetary Boundary Layer Height ($PBLH$) has been calculated.

From the analysis presented above, we deduce the following:

1. Comparisons of AMPS forecasts with radiosoundings from three representative sites (coast: McMurdo, flank: South Pole, summit: Dome C) show that the forecasts can accurately describe the trend of atmospheric meteorological parameters above the Antarctic continent, as the $R_{xy}$ for $\theta$ reached as high as 0.99 and the $R_{xy}$ for $V$ are all larger than 0.85 (Table 2).

2. We proved that the AMPS forecasts can identify the main characteristics of atmospheric turbulence over the Antarctic continent in terms of both space and time. The $R_{xy}$ of $\log_{10}(Ri)$ at MM, SP, and DC are 0.71, 0.59, and 0.53, respectively. And the AMPS can reconstruct the near-ground "convex-concave–convex" shaped $\log_{10}(Ri)$ profiles indicated by the radiosonde measurements (Fig. 5). We also find that the $R_{xy}$ of $\log_{10}(Ri)$ would be higher when the $RMSE$ of $\theta$ and $V$ are smaller (Table 2). Besides, the AMPS can better capture the trend of $\log_{10}(Ri)$ ($R_{xy}$ would be larger) at a relatively unstable atmosphere (weaker temperature inversion). Moreover, the values of $\log_{10}(Ri)$ were generally overestimated at the three sites; this is partly the result of the potential temperature gradients at the unstable atmosphere being overestimated by the AMPS, and the AMPS has generally underestimated the wind shear when it was strong.

3. The seasonal medians of the AMPS forecasts from two vertical cross-sections were presented (Figs. 8 and 9). which provides us with a broader perspective on when and where atmospheric turbulence could be highly triggered in the Antarctic region. The AMPS-forecasted $\log_{10}(Ri)$ were qualitatively verified, as its statistical distribution behaved as the expected atmospheric properties attributed by some typical large-scale phenomena or local-scale dynamics (katabatic winds, polar vortices, convection, gravity wave, etc.) over the Antarctic plateau and the ocean surrounding it. For example, a very laminar atmosphere above the Antarctic Plateau and a shallow boundary layer in the Domes area are illustrated by the AMPS forecasts.

4. The $\log_{10}(Ri)$ at the $PBLH$ were calculated and their median value is 0.316, $\log_{10}(Ri)$=0.316 in turn was used to calculate $PBLH$ and agree well with the AMPS-forecasted $PBLH$ ($R_{xy}$> 0.69). The atmosphere layer could be considered turbulent at $\log_{10}(Ri)$<0.316 (when the turbulence intensity could be comparable to that within the boundary layer).

The overall results show that the AMPS can forecast a realistic behaviour of $Ri$, and the turbulence conditions in Antarctica are well revealed; furthermore, some practical operations that want to avoid a turbulent atmosphere — such as astronomical observations (Burton, 2010), aviation safety (Gultepe and Feltz, 2019), and free space optical communication (Yin et al., 2017) — can apply the AMPS-forecasted $Ri$.

*Data availability.* The meteorological parameters measured by the radiosondes at McMurdo, South Pole that support the findings of this study are available at the Antarctic Meteorological Research Center (ftp://amrc.ssec.wisc.edu/pub), while the meteorological parameters at Dome C are available at the Antarctic Meteo-Climatological Observatory (http://www.climantartide.it). The original WRF output files of AMPS used in this study can be found at https://www2.mmm.ucar.edu/rt/amps/information/amps_esg_data_info.html.

*Video supplement.* The annual AMPS forecasts change related to the vertical cross-section through the South Pole and Dome C (Fig. 7a) is available online at https://doi.org/10.5446/60761. Another vertical cross-section through Dome A and McMurdo (Fig. 7b) is https://doi.org/10.5446/60760.

*Author contributions.* QY and XW planned the investigation; QY, XH, XW, and ZW analyzed the data; QY and YG wrote the manuscript draft; QY finished the visualizition; QY, XW, XH, XQ, and ZW performed the valiation; XW, CQ, TL, XQ, and PW reviewed and edited the

manuscript.

*Competing interests.* The authors declare that they have no conflict of interest.

*Acknowledgements.* This work was supported by the National Natural Science Foundation of China (grant Nos. 91752103, 41576185), Foundation of Key Laboratory of Science and Technology Innovation of Chinese Academy of Sciences (grant No. CXJJ-21S028) and the Foundation of Advanced Laser Technology Laboratory of Anhui Province (grant No. AHL2021QN02).

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
