# Peer review of "Antarctic atmospheric Richardson number from radiosoundings measurements and AMPS"

_Atmospheric Chemistry and Physics, 2022_

## Referee Comment (RC3)

**Antarctic atmospheric Richardson number from radiosoundings measurements and AMPS**

acp-2022-352, Qike Yang et al.

**Recommendation:** Reject

**General Comments**

This paper aims to evaluate the performance of the Antarctic Mesoscale Prediction System (AMPS) model against routine radiosonde observations from three sites across Antarctica. More specifically, the authors evaluate the differences between simulated and observed values of the reciprocal of the Richardson number, *Ri*, between March 2021 to February 2022. While this is an interesting topic and the methods appear sound, it is the opinion of this reviewer that there are issues with the scope, motivation, and interpretation of the results that would require substantial revisions to make appropriate for publication in *Atmospheric Chemistry and Physics*.

**Fatal Flaw**

The primary issue with this study is its lack of analysis or contextualization of results that warrant dissemination in a journal such as *Atmospheric Chemistry and Physics*. This issue stems from its introduction, which does not adequately frame the motivation for this study apart from generally calculating differences in Richardson number between model output and radiosonde observations. For example, there is little physical insight into the mechanisms behind these differences, how the choice of model physics affect these differences, or statistical significance for any of the computed statistics. The primary conclusions of this study seem only to comprise a report of profiles of 1/*Ri* differences between an experimental model and radiosondes, a collection of seasonal cross-sections from the model, and a brief comparison of boundary-layer height estimates. This study reads more like a technical report than a journal article with well-defined research questions based on physical mechanisms and addressing a pronounced gap in the literature. Without such analyses, it is difficult to recommend publishing in *Atmospheric Chemistry and Physics*, whose scope is "…focused on studies with important implications for our understanding of the state and behaviour of the atmosphere". Therefore, it is this reviewer's opinion that this manuscript requires too significant of changes throughout to 1) align with the scope of *ACP* and 2) provide substantial contributions to the existing literature such that the paper should be rejected with re-submission to another journal encouraged.

**Major Comments**

1. This manuscript would greatly benefit from a dedicated English language review to ensure the authors' points are properly worded and explained. There are numerous instances throughout the paper where rewording/restructuring sentences and paragraphs would highly improve the readability. Some (but not all) of these instances include lines 46—57, 195—210, 281—284, 302—304, and 383—384.

2. The motivation behind the use of 1/*Ri* instead of just *Ri* is not entirely clear beyond citing that another reference did so. One of the citations mentions that 1/*Ri* provides an easier dynamic range to plot, but in my experience this can be overcome by plotting *Ri* on a logarithmic axis to maintain the physical interpretation. Please consider changing to use *Ri* throughout or at least substantiate the use of 1/*Ri* more thoroughly.

3. In general, there is a lack of substantial analysis or interpretation of the results presented that contributes to addressing gaps in the literature. I would like to see the following addressed to improve upon this aspect of the manuscript.
    a. What are some potential underlying reasons for the discrepancies between the models and obseravtions? For example, how does the choice of model physics impact the resulting dynamic stability in simulated profiles? How does the physics in the AMPS model differ from the ECMWF analyses used by Hagelin et al. (2008)? How well does the model perform against other observations in general? How do errors depend on model forecast lead time?
    b. If the model biases can be related to parameterization schemes, what are the implications for global numerical weather prediction models? Are there underlying stable atmospheric boundary layer or marine boundary layer processes that need be more accurately accounted for in global/regional models?
    c. The monthly bulk statistics provided are informative, but could be expanded upon for emphasis. Please provide case studies from, e.g., when or where the model performed particularly well or poorly to determine if there are other large-scale phenomena biasing the model or if local-scale dynamics are not being represented properly.
    d. The determined value of critical $1/Ri$ determining the boundary layer depth should be expanded upon by a discussion of other metrics typically used to estimate the boundary layer depth. I recommend citing and discussing the work of, e.g., Pietroni et al. (2012) and Petenko et al. (2019) to better contextualize the results from section 4.2.4.
4. The linear regression analysis performed to correct model-estimated profiles of $1/Ri$ in section 4.2.2 seems interesting, albeit reads more like a calibration lab report as opposed to a journal article. Why was the form of equation (2) chosen? How does the linear regression model compare with other results in the literature? How does this regression depend on seasonality or ambient flow speed and direction? Please tie these results back to your motivation, because right now they are just presented without much practical application.
5. The vertical cross-sections in section 4.2.3 are interesting, but I am struggling to understand the value they add to the discussion on model performance versus observations. Observations do not seem to be discussed much at all in this section. Was this supposed to demonstrate the capability of the model to estimate the Richardson number close to the surface? Please tie these results back to the paper objectives of evaluating model performance.
6. Figures 2—5 are interesting, but perhaps present an information overload. I suggest condensing the figures into seasonal median plots instead of monthly medians. Please also consider utilizing a color palette that is more colorblind friendly for all figures in this manuscript (e.g., https://doi.org/10.1175/BAMS-D-13-00155.1).

**Minor and Technical Comments**
1. Line 36: please remove the break to a new paragraph here, as the content is a continuation of the sentence ending at line 35.
2. Line 40: Omit the sentence beginning "The Polar WRF model was developed…" or provide a citation instead.

3. Line 41: The sentence beginning "The simulated *Ri* by Polar WRF…" is vague and does not tell the reader much about the performance of Polar WRF in simulating turbulent exchanges of heat, momentum, etc.
4. Line 42: What specifically have the previous studies verifying the performance of AMPS find with regards to turbulence production? Please elaborate on this literature review.
5. Line 45: The sentence ending with "needs to be extended" should be elaborated upon. This may be accomplished by considering point 4 above.
6. Line 46: Please omit the phrases "And so far," and "still tremendous".
7. Line 49: The use of the word "sketchy" is too colloquial for discussion in a journal article. Please rephrase to discuss the relative lack of performance evaluations of models in Antarctica.
8. Line 54: Please elaborate on the "practical applications" mentioned at the end of point 2 to improve the motivation of this study.
9. Line 66: Please remove "or instability" to be more concise.
10. Line 91: In the sentence "The balloon scans the atmosphere," please consider replacing "scans" with "observes" or "measures" for clarity.
11. Lines 98—109: Please make it more apparent which model forecast times are selected for analysis. Please also add discussion on the choice to use forecasted fields instead of model analysis instances.
12. Line 119: Please avoid beginning a sentence with a lower case variable.
13. Lines 141—143: This information better belongs in the caption for Figures 2 and/or 3. Please consider moving.
14. Line 154: What months are being referred to when discussing the model and observation differences here?
15. Line 175: Specifically, what heights are these parameters being interpolated to? Maybe this could be depicted in a figure.
16. Lines 179—182: This paragraph discussing the use of NCL is not necessary for the presentation of results. Please instead discuss how derivatives are calculated (i.e., centered finite differencing, fitting an analytical function, etc.) and how this choice may impact the resulting profiles of $1/Ri$.
17. Line 184—185: The sentence beginning "This is because the value of $1/Ri$ can oscillate…" is vague and potentially misleading. Please clarify what is meant here.
18. Line 194: Please add a reference to a figure or citation to contextualize the discussion at the end of this paragraph.
19. Line 195: The phrase "…are afraid to conduct quantitative analysis…" is too colloquial and does not accurately portray the gaps in the literature. Please rephrase.
20. Line 210: Please provide additional discussion for why correlations are lower in the winter other than it is harder to collect observations.
21. Figure 6: This is an interesting presentation of your results, but the third dimension seems redundant to plot when the points are also colored by height. Please consider plotting instead log(1/Ri_meas) versus log(1/Ri_AMPS) in two dimensions while retaining the color shading to denote altitude. Otherwise, please discuss the results from this figure in more detail.
22. Lines 240—244: This paragraph is largely just reporting the values from table 2. Please add more substantial analysis on the resulting linear regression (see major comment 4).

23. Figure 7: From where are these terrain heights obtained? Please add information in the figure caption.
24. Figure 8: Please denote the markings of each plotted field in the figure caption.
25. Line 281—283: Please reword this sentence to better portray the importance of forecasting $1/Ri$ accurately.
26. Line 297: Please elaborate on the comparisons with sodar observations, this seems to be included without proper contextualization.

**References**

Petenko, I., S. Argentini, G. Casasanta, C. Genthon, and M. Kallistratova, 2019: Stable Surface-Based Turbulent Layer During the Polar Winter at Dome C, Antarctica: Sodar and In Situ Observations. *Boundary-Layer Meteorol*, **171**, 101–128, https://doi.org/10.1007/s10546-018-0419-6.

Pietroni, I., S. Argentini, I. Petenko, and R. Sozzi, 2012: Measurements and Parametrizations of the Atmospheric Boundary-Layer Height at Dome C, Antarctica. *Boundary-Layer Meteorol*, **143**, 189–206, https://doi.org/10.1007/s10546-011-9675-4.

Stauffer, R., G. J. Mayr, M. Dabernig, and A. Zeileis, 2015: Somewhere Over the Rainbow: How to Make Effective Use of Colors in Meteorological Visualizations. *Bulletin of the American Meteorological Society*, **96**, 203–216, https://doi.org/10.1175/BAMS-D-13-00155.1.

---

## Author Response (AR1)

**Response letter (RC1)**

Dear Editor and Reviewers:

We greatly appreciate your efforts in the previous version of the manuscript (Title: Antarctic atmospheric Richardson number from radiosoundings measurements and AMPS, Manuscript ID: acp-2022-352). We have made point-to-point responses to all the comments/suggestions raised in your review reports and made the corresponding revisions in the context. All the replies in this document are colored in blue, and the revisions/changes in the revised manuscript are marked in red.

--------Reviewer Comments--------

General comments:

The authors have addressed most of my concerns in this round. I appreciate the efforts and hard work by these authors. Nevertheless, some new problems emerge due to the substantial made to the original version, which mainly lies at the non-efficient discussion for the result interpretation. Moreover, several figures need to be modified for their unclear labels. I will recommend its publication after adequately addressing the following concerns.

**Response:**

We appreciate the Reviewer's suggestions and have revised our discussion and figures accordingly to provide more clarity on the result interpretation.

Specific comments:

Figure 1: I am confused and surprised that the authors did not correctly show the geographic coordinates for the whole study area – Antanctic continent. It is supposed to have equal latitude and longitude grid. Nevertheless, I can not see the label of latitude for this map.

**Response:**

Thanks for the Reviewer's suggestions. We have revised Figure 1 to have equal latitude and longitude grid. Latitude labels are also added for this map.

Section 2.1: The literatures on the robustness and accuracy of radiosounding measurements used here are missing, making it hard to convince the readers to believe the analysis results based on these observations. Some necessary discussion regarding this issue can be added by referring to the following references:   DOI:10.3390/rs13020173 and https://doi.org/10.5194/acp-21-17079-2021.

**Response:**

Thanks for the Reviewer's suggestions. Now we have added some discussions regarding this issue and cited the relevant references suggested by the reviewer.

**Revision in the manuscript:**

We have added

"Vaisala RS41 radiosondes have gradually replaced an older version (Vaisala RS92) starting in late 2013. These two radiosondes agree well with global average temperature differences <0.1-0.2 K in the lower stratosphere, but RS41 appears to be less sensitive than RS92 to changes in solar elevation angle (Sun et al., 2019). Besides, RS41 (1-1.5% dry bias) has better performance than RS92 (3-4% dry bias) relating to the infrared atmospheric sounding interferometer as a practical reference (Sun et al., 2021). Near-global radiosonde measurements have been used to calculate the Richardson number and derive the

boundary layer height, which is positively correlated with the results of four reanalysis products (Guo et al., 2021)."
in the revised manuscript (lines 88-94).

**References:**
Sun, B., Reale, T., Schroeder, S., Pettey, M., and Smith, R.: On the Accuracy of Vaisala RS41 versus RS92 Upper-Air Temperature Observations, Journal of Atmospheric and Oceanic Technology, 36, 635-653, 10.1175/jtech-d-18-0081.1, 2019.

Sun, B., Calbet, X., Reale, A., Schroeder, S., Bali, M., Smith, R., and Pettey, M.: Accuracy of Vaisala RS41 and RS92 Upper Tropospheric Humidity Compared to Satellite Hyperspectral Infrared Measurements, Remote Sensing, 13, 173, 10.3390/rs13020173, 2021.

Guo, J., Zhang, J., Yang, K., Liao, H., Zhang, S., Huang, K., Lv, Y., Shao, J., Yu, T., Tong, B., Li, J., Su, T., Yim, S. H. L., Stoffelen, A., Zhai, P., and Xu, X.: Investigation of near-global daytime boundary layer height using high-resolution radiosondes: first results and comparison with ERA5, MERRA-2, JRA-55, and NCEP-2 reanalyses, Atmos. Chem. Phys., 21, 17079-17097, 10.5194/acp-21-17079-2021, 2021.

Figure 7: There exists silimar issues like Figure 1.

**Response:**
    Thanks for the Reviewer's suggestions. We have revised Figure 7 by adding the latitude labels.

Figures 8 and 9: My hunch is that the labels for the lines shown in Figure 7 is not correct considering the longitude information is missing (there appears only latitude coordinate in the Abscissa, and is not enough to accurately describe the coordinates).

**Response:**
    Thanks for the Reviewer's suggestion. Now we have revised Figures 8 and 9 by adding the longitude information in the Abscissa, assisting the readers with checking the position information of the vertical cross-section.

L340: I can not find Figure 12 throughout the manuscipt.  If my guess is right, the authors may want to refer to Figures 8-9.

**Response:**
    Thanks for the Reviewer's correction. We apologize for this typo. Now we have corrected that sentence.

**Revision in the manuscript:**
    We have replaced
"Combined with the analysis of Figs. 11 and 12, the general increase in temperature-inversion strength was found to correspond to a less turbulent atmosphere with smaller $1/Ri$ (when the boundary layer is thinner), owing to large stability suppressing turbulence. "
with
"The general increase in temperature-inversion strength was considered to correspond to a less turbulent atmosphere (when the boundary layer is shallower), owing to large stability suppressing turbulence."
in the revised manuscript (lines 275-276).

In section5: The authors discussed the potential influential factors, including temperature inversion, katabatic winds and polar vortex. The nature for the latter two factors is concerned with the shear-induced turbulence. Nevertheless, the authors did not mention wind shear at all. This is supposed to be avoided. Besides, recent study (Xue et al., Q.J.2022, doi:10.1002/qj.4262) indicated that katabatic winds could be linked to trapped lee wave and result in enhanced turbulence, which can be added in attempt to increase the readability and scientific level.

Also, buoyancy, cloud cooling, convection, gravity wave (and its breaking) could be other variables that are ignored in this section. The authors can refer to the following references:

doi: 10.1088/1367-2630/aa5d63

https://doi.org/10.1016/j.envpol.2021.116534

https://doi.org/10.1029/2018JD029479

https://doi.org/10.3390/s20030677

https://doi.org/10.1088/1748-9326/abf461

doi: 10.1016/j.uclim.2022.101151

https://doi.org/10.1029/2022JD037174

https://doi.org/10.1007/s00382-021-06075-2

**Response:**

Thanks for the Reviewer's suggestion. We have conducted a more in-depth analysis of these potential physical processes in the revised manuscript, including the shear-induced turbulence (katabatic winds, polar vortices), convection (cloud cooling, boundary layer convection), temperature inversion, and the wave-induced turbulence (orographic gravity waves, trapped lee waves, inertia-gravity waves).

**Revision in the manuscript:**

We have replaced

almost all the content of Sect. 4.2.2 (Vertical cross-section)

with

[revised manuscript text omitted]

**Response letter (RC2)**

Dear Editor and Reviewers:

We greatly appreciate your efforts in the previous version of the manuscript (Title: Antarctic atmospheric Richardson number from radiosoundings measurements and AMPS, Manuscript ID: acp-2022-352). We have made point-to-point responses to all the comments/suggestions raised in your review reports and made the corresponding revisions in the context. All the replies in this document are colored in blue, and the revisions/changes in the revised manuscript are marked in red.

--------Reviewer Comments--------

General Comments

Overall, this paper is very interesting, and I think provides a benefit to the scientific community. The analysis is thorough, and the authors provide context as to how the results of the study are useful. Largely, it is well-written, with some grammatical issues, for which I provide some suggestions in the Technical Corrections section below. My biggest concern is that the authors consistently equate Richardson number to atmospheric stability, however, this is not entirely correct, as a statically stable atmosphere can still be turbulent with an Ri value below critical. If the authors re-word this discussion throughout the paper, and address my other comments, I am happy to see it published.

**Response:**

We appreciate the Reviewer's comments and have revised the entire manuscript accordingly (as detailed below), avoiding any claims that $Ri$ is directly indicative of stability.

Specific Comments

I am missing a description of why 1/Ri is useful, as opposed to simply using Ri. To me, it is odd that you would use 1/Ri as a metric for turbulence and to determine PBLH rather than simply Ri, as is more standard in previous literature. You need to provide a solid argument as to why you use 1/Ri instead, including a discussion of previous literature which supports your argument.

**Response:**

Thanks for the Reviewer's comments. We have followed your suggestion to use $Ri$ instead of $1/Ri$. However, since $Ri$ can vary by two or more orders of magnitude in the atmosphere, we have plotted it on a logarithmic axis ($\log_{10}(Ri)$, see Figs 4-6 and 8-11 in the revised manuscript) to maintain its physical interpretation. This was also suggested by Reviewer #3.

Line 64-66. This sentence implies that Ri is a direct measure or stability, but this is not true. Stability can exist in the presence of turbulence, and strong turbulence usually leads to either a neutral or unstable layer. Thus, Ri in itself does not determine static stability, though it does provide some insight into the stability of a layer. With regards to Antarctica, the PBLH is almost always either near-neutral or stable, with very few instances of instability. These near-neutral PBLHs are still turbulent, and usually, so are the stable PBLHs. Make sure here, and throughout the paper, you don't claim that a lack of turbulence (as indicated by Ri) equals a stable layer, and the presence of turbulence equals an unstable layer. Instead, contrast between turbulent and laminar, or clarify that Ri gives insight into the stability of a layer, but is not a direct measurement of stability.

**Response:**

    Thanks for the Reviewer's comments. We have revised this sentence to state that $Ri$ is estimated from the meteorological parameters measured by radiosounding, rather than a direct measurement of stability. We have also checked the entire paper to ensure that we do not claim that a lack of turbulence (or large $Ri$) implies a stable layer.

**Revision in the manuscript:**

    We have replaced

"The Richardson number ($Ri$) is a valuable parameter for atmospheric stability monitoring;"

with

"The Richardson number ($Ri$) is a valuable parameter for giving insight into atmospheric stability"

in the revised manuscript (line 17).

    We have replaced

"The measured $1/Ri$ will also be calculated using the radiosounding-measured meteorological parameters. Then, a direct comparison between measurements and forecasts can be achieved, allowing us to evaluate the ability of the AMPS to forecast atmospheric stability or instability."

with

"The radiosonde can measure meteorological parameters, which can estimate $Ri$. Using the AMPS-forecasted meteorological parameters, one also can obtain the $Ri$. Then, a comparison of $Ri$ estimated from measurements and forecasts can be achieved, allowing us to evaluate the reliability of AMPS-forecasted $Ri$ in giving insight into the atmospheric turbulence in Antarctica."

in the revised manuscript (lines 53-56).

    We have replaced

"The stability of the atmosphere can be estimated using the Richardson number ($Ri$) (Obukhov, 1971; Chan, 2008):"

with

"The Richardson number ($Ri$) is generally defined as (Richardson and Shaw, 1920; Chan, 2008):"

in the revised manuscript (lines 113).

    We have replaced

"To evaluate the performance of the AMPS in forecasting atmospheric stability or instability over the Antarctic continent, the $1/Ri$ forecasted by AMPS and measured by radiosoundings will be provided."

with

"To evaluate the performance of AMPS in forecasting the possibility of triggering turbulence over the Antarctic continent, the $Ri$ estimations between radiosoundings and AMPS forecasts will be compared."

in the revised manuscript (lines 160-161).

    We have added

"Importantly, it should be noted that it is the range of turbulence (or $PBLH$) that would be suppressed by the temperature inversion and the turbulence intensity could be strong within the inversion layer (Petenko et al., 2019)"

in the revised manuscript (lines 279-281).

    We have replaced

"$1/Ri_{PBLH}$ would thus be a helpful reference standard for judging whether an atmospheric layer is stable ($1/Ri < 1/Ri_{PBLH}$) or unstable ($1/Ri > 1/Ri_{PBLH}$)."

with

"since the value of $Ri_{PBLH}$ is a helpful reference for judging whether atmospheric turbulence is likely to be suppressed ($Ri > Ri_{PBLH}$) or developed ($Ri < Ri_{PBLH}$)."
in the revised manuscript (lines 311-312).

We have replaced

"Further, the $Ri_{PBLH}$ forecasted by the AMPS was employed to understand how to evaluate atmospheric stability or instability using the value of $1/Ri$."
with

"Finally, the $\log_{10}(Ri)$ at the Planetary Boundary Layer Height ($PBLH$) has been calculated."
in the revised manuscript (lines 344-345).

We have replaced

"Finally, the $1/Ri$ at the planetary boundary layer height ($PBLH$), $1/Ri_{PBLH}$, has been provided as a reference standard for judging atmospheric stability."
with

"The $\log_{10}(Ri)$ at the $PBLH$ were calculated and their median value is 0.316, $\log_{10}(Ri)$=0.316 in turn was used to calculate $PBLH$ and agree well with the AMPS-forecasted $PBLH$ ($R_{xy} > 0.69$). The atmosphere layer could be considered turbulent at $\log_{10}(Ri)$ <0.316 (when the turbulence intensity could be comparable to that within the boundary layer)."
in the revised manuscript (lines 364-366).

Line 68. Please clarify bullet point #3. Particularly, it is unclear to me what you mean when you say the analysis was extended for three sites to two vertical cross-sections at a high horizontal resolution. More description of what this means is necessary.

**Response:**

Thanks for the Reviewer's comments. We have provided more details on this point. The results of two vertical cross-sections for $Ri$, we believe, enable readers to better understand the turbulence conditions in both vertical and horizontal dimensions, instead of only focusing on the vertical dimension (or atmospheric column; e.g., Hagelin et al. 2008 and Geissler and Masciadri 2006).

**Revision in the manuscript:**

We have replaced

"3. We also extended the analysis of $1/Ri$ done by (Hagelin et al., 2008) for three sites to two vertical cross-sections at a high horizontal resolution. Finally, regions and periods that are favourable for triggering atmospheric turbulence (or instability) can be identified. Thus, this study provides a better perspective on atmospheric dynamics in Antarctica."
with

"3. Two vertical cross-sections for $Ri$ will be given, which may provide a better perspective on the turbulence conditions in both vertical and horizontal dimensions, instead of only focusing on the vertical dimension (or atmospheric column; e.g., Hagelin et al. 2008 and Geissler and Masciadri 2006). This will help to identify regions and periods that are favorable for triggering atmospheric turbulence in Antarctica. Moreover, this will enable us to correlate the $Ri$ distribution with some large-scale phenomena or local-scale dynamics (katabatic winds, polar vortices, convection, gravity wave, etc.) in Antarctica, and the underlying physical processes of Antarctic atmospheric turbulence will be investigated."
in the revised manuscript (lines 58-63).

**References:**

Hagelin, S., Masciadri, E., Lascaux, F., and Stoesz, J.: Comparison of the atmosphere above the South Pole, Dome C and Dome A: first attempt, Monthly Notices of the Royal Astronomical Society, 387, 1499-1510, 10.1111/j.1365-2966.2008.13361.x, 2008.

Geissler, K. and Masciadri, E.: Meteorological Parameter Analysis above Dome C Using Data from the European Centre for Medium-Range Weather Forecasts, Publications of the Astronomical Society of the Pacific, 118, 1048-1065, 10.1086/505891, 2006.

Line 11 and throughout paper. Ri should be calculated using potential temperature and wind speed, not temperature, which you show with Eq. 1. Make sure to specify in the text throughout the paper that Ri is calculated using potential temperature.

**Response:**

   Thanks for the Reviewer's comments. We have examined the entire paper to ensure that we clarify that $Ri$ is calculated using potential temperature.

**Revision in the manuscript:**

   We have replaced

"The Antarctic atmospheric $Ri$, calculated using the temperature and wind speed"

with

"The Antarctic atmospheric $Ri$, calculated from the potential temperature and wind speed"

in the revised manuscript (lines 3).

   We have replaced

"as it can be calculated by the routine meteorological parameters (temperature and wind speed)."

with

"as it can be calculated from the routine meteorological parameters (potential temperature and wind speed)."

in the revised manuscript (lines 36-37).

   We have replaced

"To carry out a detailed comparison of the temperature and wind speed (on which $1/Ri$ depends)"

with

"To carry out a detailed comparison of potential temperature and wind speed (on which $Ri$ depends)"

in the revised manuscript (line 48).

   We have replaced

"4.1 Temperature and wind speed."

with

"4.1 Potential temperature and wind speed"

in the revised manuscript (line 126).

   We have replaced

"Figure 2. The monthly median for temperature forecasted by the AMPS (solid lines) and temperature difference"

with

"Figure 2. The seasonal median of potential temperature ($\theta$) estimated by the radiosonde measurements (solid lines) and potential temperature difference ($\Delta\theta$)"

in the revised manuscript (see the caption for Figure 2, line 154, page 7).

We have replaced

"Fig. 2 shows that the forecasted temperature profiles of SP and DC in the first 5 km are similar, 15-20 K colder than the MM. The median difference in the temperature is of the order of 1 K in the high part of the atmosphere (see filled areas in Fig. 2). However, in proximity to the ground, the median difference becomes more significant, especially for winter (similar to Bromwich et al., 2013), at more than 4 K near the ground at SP."

with

"Fig. 2 shows that the median difference for $\theta$ is of the order of 1 K in the first 5 km (except for the atmosphere layer in proximity to the ground). Above 5 km, the AMPS has obviously underestimated the at MM, while the forecasts at SP and DC are closer to measurements."

in the revised manuscript (lines 148-150).

We have replaced

"In summary, the AMPS almost well forecasted the temperature and wind speed"

with

"It seems the AMPS can well capture the trend of $\theta$ and $V$ as the correlation coefficient ($R_{xy}$) are all larger than 0.84."

in the revised manuscript (lines 156-157).

We have replaced

"However, the median differences are not similar to those of the previous temperature and wind speed."

with

"However, the median differences are not presented like the $\theta$ and $V$."

in the revised manuscript (lines 168-169).

We have replaced

"This includes quantifying the accuracy of meteorological parameters (Temperature and wind speed, on which the $1/Ri$ depends),"

with

"This includes evaluating the accuracy of meteorological parameters ($\theta$ and $V$, on which the $Ri$ depends),"

in the revised manuscript (lines 341-342).

Sect. 2.1. Add the frequency with which the radiosondes were launched. I see that you use data from March 2021 to February 2022, but how often within that year were radiosondes launched? Does this differ for the different sites?

**Response:**

Thanks for the Reviewer's comments. We have added the information on the frequency of radiosonde launched within that year. Generally, the radiosonde was launched once a day at the same hour (sometimes twice a day at MM and SP). In total, 518, 508, and 340 profiles were available at MM, SP, and DC from March 2021 to February 2022.

**Revision in the manuscript:**

We have replaced

"Generally, the radiosonde was launched once a day at the same hour (sometimes twice a day at MM and SP). In total, 518, 508, and 340 profiles were available at MM, SP, and DC from 2021 March to 2022 February."
in the revised manuscript (lines 83-85).

Sect 2.1. Did you remove questionable radiosonde measurements near the surface? Usually, near-surface radiosonde measurements are unreliable due to the radiosonde needing time to equilibrate to the outside temperatures once it is launched, so the temperature values reflect warmer values than reality. Thus, below about 20 m, radiosonde measurements are usually unreliable, and make it look like the atmosphere is unstable, when it really isn't. Make sure you are taking this into account.

**Response:**

Thanks for the Reviewer's comments. We did have removed the near-ground radiosonde measurements. However, it was the first 10 m data that were discarded. This is because, the first AMPS eta grid is ~10 m above the ground, and the radiosonde measurements were linearly interpolated to the same height for comparison. We have added comments on this issue in the revised manuscript.

**Revision in the manuscript:**

We have added

"On the other hand, it should be noted that the near-surface radiosonde measurements could be less reliable, as it was just released from the operator's hand (or some machine). Hagelin et al. (2008) conclude that the radiosoundings are ~1 K colder than the Automatic Weather Station at Dome C and ~2 K at the South Pole. In this study, the radiosonde measurements in the first ~10 m above the ground were not used. This is also because the first AMPS grid is ~10 m above the ground."
in the revised manuscript (lines 135-139).

**References:**

Hagelin, S., Masciadri, E., Lascaux, F., and Stoesz, J.: Comparison of the atmosphere above the South Pole, Dome C and Dome A: first attempt, Monthly Notices of the Royal Astronomical Society, 387, 1499-1510, 10.1111/j.1365-2966.2008.13361.x, 2008.

Figure 1. Add latitude values. Also, in this figure caption is the first time you introduce Dome A. You should introduce the significance of the Dome A site earlier on in the introduction or methods.

**Response:**

Thanks for the Reviewer's suggestions. We have added the latitude values in Fig. 1. And we have introduced the significance of the Dome A site.

**Revision in the manuscript:**

We have added

"Dome A (DA) is also marked in Fig. 1, which is the highest location (4083 m) on the Antarctic plateau, and the atmospheric conditions above it will also be analyzed in this study (Sect. 4.2.3)."
in the revised manuscript (lines 80-81).

Line 116. You need to be careful with statements like this. As mentioned in an earlier comment, Ri does not directly determine atmospheric stability. Instead, the change in potential temperature with altitude is a direct indicator of stability (see Sect. 2.2 of https://amt.copernicus.org/articles/15/4001/2022/amt-15-4001-2022.html). A layer can have an increase of potential temperature with altitude (indicative of a

stable layer), but wind shear still provides some turbulence, and Ri is below the critical value. Change your wording here and throughout the paper to avoid claiming that Ri is directly indicative of stability.

**Response:**

Thanks for the Reviewer's suggestions. We have revised this statement and checked the entire paper to avoid claiming that $Ri$ is directly indicative of stability (as addressed in the comment regarding Lines 64-66).

**Revision in the manuscript:**

We have replaced

"The stability of the atmosphere can be estimated using the Richardson number ( $Ri$ ) (Obukhov, 1971; Chan, 2008):"

with

"The Richardson number ( $Ri$ ) is generally defined as (Richardson and Shaw, 1920; Chan, 2008):"

in the revised manuscript (line 113).

Line 121. As I previously stated, the production of turbulence does not always equate to an unstable atmosphere. A truer statement is that buoyantly driven turbulence often leads to instability, but mechanically driven turbulence often does not lead to instability. In Antarctica, the turbulence is dominated by mechanical rather than buoyant forcings, so I would not make this claim.

**Response:**

Thanks for the Reviewer's suggestions. We have revised the statements and removed the words "unstable atmosphere"

**Revision in the manuscript:**

We have replaced

"The production of atmospheric turbulence (or unstable atmosphere) was shown to be tightly correlated with the $Ri$ ."

with

"The development of atmospheric turbulence was shown to be tightly correlated with the $Ri$ ."

in the revised manuscript (line 119).

Line 126. Why does 1/Ri provide better evidence of atmospheric stability? 1/Ri would provide the same evidence as would Ri. Is it because 1/Ri may be easier to interpret, since larger 1/Ri means more turbulence? Clarify this.

**Response:**

Thanks for the Reviewer's suggestions. We have followed your suggestion and used $Ri$ instead of $1/Ri$ . Nevertheless, we have plotted it on a logarithmic axis ( $\log_{10}(Ri)$ ), since $Ri$ can vary massively in the atmosphere (by two or more orders of magnitude). Using $\log_{10}(Ri)$ was also suggested by Reviewer #3.

The reason for using $1/Ri$ in the original manuscript was suggested by Geissler, K., (2006), as they think that $1/Ri$ provide readers with better evidence of stability differences in different months from a visual point of view.

**Revision in the manuscript:**

We have replaced

"In this study, the inverse of the Richardson number ($1/Ri$) was used to provide better evidence of atmospheric stability, as in Geissler and Masciadri (2006) and Hagelin et al. (2008). The larger the $1/Ri$, the higher the probability of triggering turbulence in the atmosphere."

with

"In the results of this study, the logarithm of $Ri$, $\log_{10}(Ri)$, is presented instead of $Ri$ itself, because $Ri$ can vary by two or more orders of magnitude in the atmosphere."

in the revised manuscript (lines 123-124).

**References:**

Geissler, K. and Masciadri, E.: Meteorological Parameter Analysis above Dome C Using Data from the European Centre for Medium-Range Weather Forecasts, Publications of the Astronomical Society of the Pacific, 118, 1048-1065, 10.1086/505891, 2006.

Sect 3. Please explain how 1/Ri profiles were calculated. Do you calculate this at each altitude by comparison with the lowest measurement? Or do you calculate it between consecutive vertical levels?

**Response:**

We think the $Ri$ was calculated between consecutive vertical levels, as a centered finite difference operation was performed on the height dimension to estimate the gradient in Eq. (1). This is consistent with the NCL code computing Ri (https://www.ncl.ucar.edu/Document/Functions/Contributed/rigrad_bruntv_atm.shtml), we have checked the NCL code (a file named "contributed.ncl") and found it performs a centered finite difference operation (using a function called "center_finite_diff_n").

**Revision in the manuscript:**

We have added

"To calculate $Ri$, a centered finite difference operation was used to estimate the gradient in Eq. (1)."

in the revised manuscript (lines 117-118).

Fig. 2. Since potential temperature, rather than temperature, is the variable that directly feeds into calculating Ri, I would recommend plotting this figure with potential temperature rather than temperature. Additionally, potential temperature is a more standard and helpful variable used to visual atmospheric stability.

**Response:**

Thanks for the Reviewer's suggestions. We have replaced the plots of temperature with potential temperature in Fig. 2 and revised the corresponding descriptions in the text.

**Revision in the manuscript:**

We have replaced

"Fig. 2 shows that the forecasted temperature profiles of SP and DC in the first 5 km are similar, 15-20 K colder than the MM. The median difference in the temperature is of the order of 1 K in the high part of the atmosphere (see filled areas in Fig. 2). However, in proximity to the ground, the median difference becomes more significant, especially for winter (similar to Bromwich et al., 2013), at more than 4 K near the ground at SP."

with

"Fig. 2 shows that the median difference for $\theta$ is of the order of 1 K in the first 5 km (except for the atmosphere layer in proximity to the ground). Above 5 km, the AMPS has obviously underestimated the at MM, while the forecasts at SP and DC are closer to measurements."

Line 165-168. Do you have any evidence to support this hypothesis? Because, with your reasoning, the opposite theory could also be formed. Make sure you support this hypothesis with evidence, otherwise remove it.

**Response:**

Thanks for the Reviewer's comments. We have removed this statement, as we have no sufficient evidence to support this hypothesis yet.

**Revision in the manuscript:**

We have removed

"Nevertheless, it should be noted that both the measurement accuracy and position of radiosoundings may be affected by their flight, and the difference may be smaller when compared with a fixed measuring instrument, such as the Automatic Weather Station (Hagelin et al., 2008). In other words, the functional performance of the AMPS may be better than the results shown in Figs. 2 and 3."
in the original manuscript (lines 165-168).

More examples of issue directly inferring stability from Ri:
Line 171, Line 186, Line 214, Line 291-292, Line 358, Line 376-377, Line 379-380

**Response:**

Thanks for the Reviewer's comments. We have revised all these sentences to avoid claiming that $Ri$ is directly indicative of stability.

**Revision in the manuscript:**

We have replaced

"To evaluate the performance of the AMPS in forecasting atmospheric stability or instability over the Antarctic continent, the $1/Ri$ forecasted by AMPS and measured by radiosoundings will be provided."
with
"To evaluate the performance of AMPS in forecasting the possibility of triggering turbulence over the Antarctic continent, the $Ri$ estimations between radiosoundings and AMPS forecasts will be compared."
in the revised manuscript (lines 160-161).

We have replaced

"Nevertheless, the AMPS-forecasted $Ri$ can identify that the atmosphere above MM tends to be more unstable (or turbulent, $1/Ri$ is large) than SP and DC"
with
"In Fig. 4, one can see that the AMPS-forecasted $Ri$ can identify that the atmosphere above MM tends to be more turbulent ($Ri$ is smaller) than SP and DC."
in the revised manuscript (lines 173-174).

We have removed

"Fig. 5c shows that $RMSE$ at MM is the largest, which may be due to the relatively unstable atmosphere above it and $1/Ri$ could fluctuate massively."
in the original manuscript (lines 214-215).

We have replaced

"$1/Ri_{PBLH}$ would thus be a helpful reference standard for judging whether an atmospheric layer is stable ($1/Ri < 1/Ri_{PBLH}$) or unstable ($1/Ri > 1/Ri_{PBLH}$)."

with

"since the value of $Ri_{PBLH}$ is a helpful reference for judging whether atmospheric turbulence is likely to be suppressed ($Ri > Ri_{PBLH}$) or developed ($Ri < Ri_{PBLH}$)."

in the revised manuscript (lines 311-312).

We have replaced

"Further, the $PBLH$ forecasted by the AMPS was employed to understand how to evaluate atmospheric stability or instability using the value of $1/Ri$."

with

"Finally, the $\log_{10}(Ri)$ at the Planetary Boundary Layer Height ($PBLH$) has been calculated."

in the revised manuscript (lines 344-345).

We have removed

"we find that unstable atmospheres are likely to be triggered over the ocean, move toward the Antarctic Plateau, and become stable."

in the original manuscript (lines 376-377).

We have replaced

"which could be a helpful reference standard for judging whether the atmospheric layer is stable ($1/Ri < 1/Ri_{PBLH}$) or unstable ($1/Ri > 1/Ri_{PBLH}$) when using AMPS-forecasted $1/Ri$"

with

"The atmosphere layer could be considered turbulent at $\log_{10}(Ri_{PBLH}) < 0.316$ (when the turbulence intensity could be comparable to that within the boundary layer)."

in the revised manuscript (lines 365-366).

Fig. 4. Why do you show the median profiles from the observations here, but you show the difference between the model and observations for Fig. 2 and 3. I would try to remain consistent, but otherwise give an explanation as to why you show them differently.

**Response:**

The $Ri$ can vary by several orders of magnitude in the atmosphere, which differs significantly from the cases of potential temperature and wind speed. Given this, at the beginning, we attempt to examine whether AMPS can reconstruct an accurate shape of $Ri$ profile (while median difference is not suitable for this purpose). And we find that the AMPS can well reconstruct the near-ground "convex-concave–convex" shaped $\log_{10}(Ri)$ profiles indicated by the radiosonde measurements (see more details in Fig. 5 in the revised manuscript). Nevertheless, we have presented the $Ri$ difference between the model and observations in Table 2 and Fig. 6 in the revised manuscript, and discuss the model errors in detail.

**Revision in the manuscript:**

We have replaced

"However, the median differences are not similar to those of the previous temperature and wind speed. This is because, the value of $1/Ri$ can oscillate massively in the atmosphere, and precise quantification seems less plausible."

with

"However, the median differences are not presented like the $\theta$ and $V$. This is because, the $Ri$ value can vary massively (by two or more orders of magnitude) in the atmosphere, and a precise quantification seems less plausible. Considering this, we initially intended to examine whether AMPS can reconstruct an accurate shape of $\log_{10}(Ri)$ profile (while median difference is not suitable for this purpose), and the results from radiosoundings and AMPS forecasts are both presented. Nevertheless, the model bias is by all means of great significance, and it will be discussed later (see Table 2, and Fig. 6)."
in the revised manuscript (lines 168-173).

Fig. 6. This is difficult to interpret. Please provide more discussion of what is shown in this figure, and what the significance is.

**Response:**
   We have plotted a new figure to replace the old one as it is hard to interpret. The new figure (see Fig. 6 in the revised manuscript) demonstrates how model errors depend on the atmospheric conditions: potential temperature gradient and wind shear.
   The old Fig. 6 (in the original manuscript) show us the distributions of $Ri$ between radiosoundings and AMPS forecasts at all height levels, illustrating the discrepancies between the models and observations.

**Revision in the manuscript:**
   We have replaced
"4.2.2 Vertical distribution
To further investigate the difference between the radiosounding and model results, the vertical distributions of the whole-year $\log_{10}(1/Ri)$ at the three sites are shown in Fig. 6. The reason for using $\log_{10}(1/Ri)$ (not $1/Ri$, as presented in Sect. 4.2.1) is that we find that the distribution will be more linear at the same height. Fig. 6a shows that the data points are distributed mainly over the diagonal line between the measurements and the forecasts, this suggests the AMPS can make a better forecast $\log_{10}(1/Ri)$ at McMurdo than at the South Pole (Fig. 6b) and Dome C (Fig. 6c).

[Figure]

Figure 6. the vertical distributions of whole-year $\log_{10}(1/Ri)$ from the AMPS and the measurements at McMurdo (a), the South Pole (b), and Dome C (c). The coloured grid lines represent the fitted planes using the presented data points.

   We assume that the difference between the measurements and model results was correlated with height (or AGL, here indicated by $H$), and a simple model to improve the AMPS-forecasted $1/Ri$ by considering $H$ (km) is given by the following linear equation:

$$\log_{10}\left(1/Ri_{\text{ImpAMPS}}\right)=C_1+C_2\cdot\log_{10}\left(1/Ri_{\text{AMPS}}\right)+C_3\cdot H . \qquad (1)$$

The three undetermined coefficients ($C_1$, $C_2$, and $C_3$) can be determined using a linear fitting by the measurements, $1/Ri_{\text{Mea.}}$ (i.e. $1/Ri_{\text{ImpAMPS}}$ will be replaced with $1/Ri_{\text{Mea.}}$ for fitting).

Table 2 lists the fitted model and its corresponding improvements. The correlation coefficient at all three sites increased after using the fitted model, which means that the fitted model can be used to modify the AMPS forecasts ($1/Ri_{\text{AMPS}}$) and obtain an improved result ($1/Ri_{\text{ImpAMPS}}$). Through analysis of the fitted model, one can see the undetermined coefficient $C_2$ is less than 1, we find this is due to the AMPS underestimated $\log_{10}\left(1/Ri\right)$ when $\log_{10}\left(1/Ri\right)<0$; the undetermined coefficient $C_3$ is negative, indicating the negative regulating action for $1/Ri_{\text{AMPS}}$ with increasing height.

Table 2. The fitted model for improving the AMPS-forecasted $1/Ri$.

| Site | Model: $\log_{10}\left(1/Ri_{\text{ImpAMPS}}\right)=C_1+C_2\cdot\log_{10}\left(1/Ri_{\text{AMPS}}\right)+C_3\cdot H$ | $R_{xy}$ (Before)[a] | $R_{xy}$ (After)[b] |
|---|---|---|---|
| McMurdo | $\log_{10}\left(1/Ri_{\text{ImpAMPS}}\right)=0.1222+0.4826\cdot\log_{10}\left(1/Ri_{\text{AMPS}}\right)-0.0660\cdot H$ | 0.7215 | 0.7691 |
| South Pole | $\log_{10}\left(1/Ri_{\text{ImpAMPS}}\right)=-0.4414+0.3827\cdot\log_{10}\left(1/Ri_{\text{AMPS}}\right)-0.0387\cdot H$ | 0.6009 | 0.6391 |
| Dome C | $\log_{10}\left(1/Ri_{\text{ImpAMPS}}\right)=-0.3596+0.3977\cdot\log_{10}\left(1/Ri_{\text{AMPS}}\right)-0.0457\cdot H$ | 0.4730 | 0.5235 |

[a] Correlation coefficient between $\log_{10}\left(1/Ri_{\text{Mea.}}\right)$ and $\log_{10}\left(1/Ri_{\text{AMPS}}\right)$ (i.e. before using the fitted model).

[b] Correlation coefficient between $\log_{10}\left(1/Ri_{\text{Mea.}}\right)$ and $\log_{10}\left(1/Ri_{\text{ImpAMPS}}\right)$ (i.e. after using the fitted model)."

with
"

[Figure]

Figure 6. Performance of the AMPS under different atmospheric conditions. (a) and (b) are respectively the case of potential temperature gradient ($G=\partial\theta/\partial z$) and wind shear ($S=\left[\left(\partial u/\partial z\right)^2+\left(\partial v/\partial z\right)^2\right]^{1/2}$). $G$ (color of the bin in (a)), $S$ (color of the bin in (b)), $\Delta G$ (stem above the bin in (a)) and $\Delta S$ (stem above the bin in (b)) are presented using the median value for each $0.5\times0.5$ bin of $\log_{10}\left(Ri\right)$.

Fig. 6 shows the AMPS performance under different potential temperature gradient ($G=\partial\theta/\partial z$) and wind shear ($S=\left[\left(\partial u/\partial z\right)^2+\left(\partial v/\partial z\right)^2\right]^{1/2}$). The statistical results presented in Fig. 6 were counted based

on all the collected data points at the three sites (MM, SP, and DC) for an entire year. One can see that the $Ri$ was overestimated by the AMPS at unstable atmosphere (see light blue bin in Fig. 6a), where the AMPS has overestimated the potential temperature gradient (i.e. $\Delta G > 0$). But for strong temperature inversion (see dark red bin in Fig. 6a), the AMPS has underestimated the $G$ and $Ri$. As for strong wind shear conditions (see dark red bin in Fig. 6b), when the $Ri$ is small (basically corresponding to a near-surface layer with a high probability of triggering strong turbulence, as in Fig. 4), the AMPS has underestimated the intensity of wind shear ($\Delta S < 0$). This may be caused by the AMPS has underestimated the wind speed near the ground (as in Fig. 3). In sum, if the model aims for a more accurate forecast of $Ri$, the biases under these atmospheric conditions need to be corrected."
in the revised manuscript (lines 217-225).

Fig. 7. It is not necessary to include the color bar twice. This would look neater if you just had one centrally located color bar. Also, add latitude numbers.

**Response:**
   In Fig. 7, we have substituted the two colors bars with a single color bar located at the center. Moreover, we have included the values of latitude for the map.

Fig. 8. It may not be necessary to explain how wind barbs are interpreted, as this is standard practice in meteorology. Additionally, I again suggest just showing the color bar once, to clean up the figure a little bit.

**Response:**
   We have removed the explanations about the wind barbs. Additionally, we have replaced the four colors bars with one centrally located color bar in Figs. 8 and 9. Also, we have added the values of longitude (as there are only latitudes in the original figure).

Line 281-283. Give some examples of why one might want to know when the atmosphere is turbulent in Antarctica in order to avoid it. For example, aviation, certain scientific activities, etc.

**Response:**
   We have added some examples that want to avoid atmospheric turbulences (astronomical observatons, aviation safety).

**Revision in the manuscript:**
   We have replaced
"Thus, forecasting $1/Ri$ is necessary if one wishes to avoid a turbulent atmosphere in Antarctica,."
with
"Some activities in Antarctica require a non-turbulent atmosphere, such as astronomical observations (Burton, 2010) and aviation safety (Gultepe and Feltz, 2019)."
in the revised manuscript (lines 301-302).

Line 351. Explain why a turbulent atmosphere over the ocean is expected. Oceanic waves? Heat flux from open ocean or sea ice leads?

**Response:**
   We have elucidated the reasons for the expected turbulent atmosphere over the ocean, based on the analysis of boundary layer convection (heat flux from open ocean) and polar vortices (intense zonal winds). The oceanic waves are generally induced by wind blowing along the air-water interface. thus we

would like to explain it from the perspective of wind (instead of oceanic waves), and the readers can check it out from the wind fields in Figs. 8 and 9. There, the strong wind over the ocean corresponds to polar vortices.

**Revision in the manuscript:**

We have added

"Boundary layer convection is generated by forcing from the ground, solar heating of the ground during sunny days causes thermals of warmer air to rise and convection will form (He et al., 2020), then the turbulence could be developed forced by buoyancy (Verma et al., 2017). The albedo of fresh snow over sea ice is very high, while that for open water is relatively small (Hines et al., 2015). Thus, solar heating will be much more stand out over open water and lead to the thermal convection boom. This can be used to reasonably explain the results in Fig. 8, that the $Ri$ over ocean (A2 area) is smaller than over ice shelf (A6 area), and the A5 area can be regarded as a "transition region" (sea ice and open water could both exist) between them with an intermediate value of $Ri$ ."
in the revised manuscript (lines 266-272).

We have added

"Moreover, the strongest zonal winds are located over the ocean (Zuev and Savelieva, 2019b), the interaction between the zonal wind and the ocean surface may generate wind shear and facilitate the development of turbulence (see areas A2 and A12 in Fig. 8, B2 and B9 in Fig. 9)."
in the revised manuscript (lines 256-558).

Line 384. Give more examples of why these results will be useful to others, aside from astronomy. Aviation is an important one.

**Response:**

We have added the example: aviation safety.

**Revision in the manuscript:**

We have replaced

"The overall results show that the AMPS can forecast the behaviour of $1/Ri$ and could be applied to practical operations such as astronomy in Antarctica, which is interested in the impacts of atmospheric turbulence (Burton, 2010)."
with

"The overall results show that the AMPS can forecast a realistic behaviour of $Ri$ , and the turbulence conditions in Antarctica are well revealed; furthermore, some practical operations that want to avoid a turbulent atmosphere — such as astronomical observations (Burton 2010), aviation safety (Gultepe and Feltz 2019) and free space optical communication (Jianjun Yin et al. 2017) — can apply the AMPS-forecasted $Ri$ ."
in the revised manuscript (lines 367-370).

Technical Corrections
Line 11. The statement in parentheses is not grammatically correct. Consider "Ri; a critical parameter which determines the possibility that turbulence could be triggered."

**Revision in the manuscript:**

We have replaced
"( $Ri$ ; a critical parameter determining the possibility of turbulence could be triggered)."
with

"($Ri$; a valuable parameter which determines the possibility that turbulence could be triggered)."
in the revised manuscript (line 2).

Line 17. Instead of "gains" perhaps use the word "increases."

**Revision in the manuscript:**
   We have replaced
"where the performance gains during the warm seasons."
with
"The $Ri$ was generally underestimated by the AMPS and the AMPS could better capture the trend of $\log_{10}(Ri)$ under relatively unstable atmospheric conditions."
in the revised manuscript (lines 8-9).

Line 46. It is improper to start a sentence with the word "And". Remove this word.

**Revision in the manuscript:**
   We have replaced
"And so far"
with
"Presently"
in the revised manuscript (lines 35).

Line 46-47. The sentence would read better as "Presently, monitoring a wide range of atmospheric turbulence over the Antarctic continent is tremendously difficulty, while the atmospheric Richardson number…"

**Revision in the manuscript:**
   We have replaced
"And so far, monitoring a wide range of atmospheric turbulence over the Antarctic continent is still tremendous difficult at present, while the atmospheric Richardson number…"
with
"Presently, monitoring a wide range of atmospheric turbulence over the Antarctic continent is tremendously difficult, but atmospheric $Ri$ …"
in the revised manuscript (lines 35-36).

Line 47-48. The statement in parentheses is again improper grammar. See my comment on Line 11 above and apply a similar fix here.

**Response:**
   It appears that such statements have been reiterated in the manuscript, so we have deleted the statement in parentheses.

**Revision in the manuscript:**
   We have replaced
" while the atmospheric Richardson number ($Ri$; a critical parameter judging the possibility of the turbulence could be triggered) is easier to obtain"
with
"but the atmospheric $Ri$ is easier to obtain"
in the revised manuscript (line 35-36).

Line 48. Should be "calculated from" instead of "calculated by"

**Revision in the manuscript:**
We have replaced
"as it can be calculated by the routine meteorological parameters"
with
"as it can be calculated from the routine meteorological parameters"
in the revised manuscript (lines 36).

Line 49. What do you mean by "sketchy?" Use a more descriptive word or statement here. The word "sketchy" is too colloquial for a scientific journal.

**Revision in the manuscript:**
We have replaced
"a precise evaluation of atmospheric models to forecast atmospheric $Ri$ in Antarctica is sketchy."
with
"few studies have evaluated atmospheric models to forecast $Ri$ in Antarctica,"
in the revised manuscript (lines 37).

Line 52. Would read better as "However, their research has some specific shortcomings (or problems that need further study)…"

**Revision in the manuscript:**
We have replaced
"However, their researches have some specific shortages (or problems that need further study)"
with
"However, their research has some specific shortcomings (or problems that need further study)"
in the revised manuscript (lines 41-42).

Line 80. It would be more appropriate to say "Sect. 6 summarizes the main findings and primary takeaways of this study."

**Revision in the manuscript:**
We have replaced
"Sect. 6 summarises the main aspects of this study."
with
"Sect. 5 summarizes the main findings and primary takeaways of this study."
in the revised manuscript (lines 72-73).

Line 92-93. It is not necessary to say "depending on the ascent speed" if you don't give a range of what the ascent speed could be. Instead, comment that with a typical ascent rate of 5 m/s, and a logging frequency of 1 Hz, the vertical resolution is approximately 5 m.

**Revision in the manuscript:**
We have replaced
"The balloon scans the atmosphere between the ground and an altitude of 10–25 km (low in winter and high in summer); the vertical resolution is approximately 5 m, depending on the ascent speed."
with

"In Antarctica, the radiosondes measure the atmosphere between the ground and an altitude of 10–25 km (low in winter and high in summer) with a typical ascent rate of 5 $\mathrm{m\ s^{-1}}$, and a logging frequency of 1 Hz; then the vertical resolution is approximately 5 m."
in the revised manuscript (lines 95-97).

Line 132-133. Not necessary to again state the time over which the analysis is done, as this is already stated in the methods.

**Revision in the manuscript:**
    We have removced
"The AMPS forecasts and radiosoundings used for this comparison were obtained from March 2021 to February 2022."
in the original manuscript (lines 132-133).

Line 195. This wording is not ideal. I don't think you should call other researchers "afraid." Please rephrase to explain why this analysis has not been done before.

**Revision in the manuscript:**
    We have replaced
"Some researchers are afraid to conduct quantitative analysis for the model-estimated $1/Ri$ as it always varies dramatically"
with
"Quantitative analysis for the estimated $Ri$ from the numerical models was generally missed as it always varies dramatically (e.g., Hagelin et al., 2008, who focused on the qualitative analysis)"
in the revised manuscript (lines 184-185).

Line 199. Say "Additionally" instead of "Besides." The word "Besides" does not make sense in this context.

**Response:**
    We have removed the sentence "Besides, all profile data meeting the time constraints are also limited to a value range of (0, 1)," in the revised manuscript, as we have used $\log_{10}(Ri)$ instead of $Ri$ (as mentioned in the 1st response to the Specific Comments). Therefore, the limitation for the value range was not necessary since $\log_{10}(Ri)$ was used.

**Revision in the manuscript:**
    We have removed
"Besides, all profile data meeting the time constraints are also limited to a value range of (0, 1),"
in the original manuscript (lines 199-200).

Line 222. Say "which suggests" instead of "this suggests."

**Response:**
    We have removed the sentence "this suggests the AMPS can make a better forecast" in the revised manuscript, as the results in this section are hard to interpret (as addressed in Specific Comments regarding Fig. 6).

**Revision in the manuscript:**
    We have removed

"this suggests the AMPS can make a better forecast"
in the original manuscript (lines 222-223).

Line 341. Say "shallower" instead of "thinner." (Also apply this to Line 372).

**Revision in the manuscript:**
   We have replaced
"when the boundary layer is thinner"
with
"when the boundary layer is shallower"
in the revised manuscript (line 276).
   We have replaced
"a thin boundary layer"
with
"a shallow boundary layer"
in the revised manuscript (line 363).

Line 348. Say "laminar" instead of "calm." (Also apply this to Line 372).

**Revision in the manuscript:**
   We have replaced
"One can also see that the free atmosphere in Antarctica becomes relatively calm during summer,"
with
"which corresponds to the relatively turbulent free atmosphere with low $Ri$ values over the ocean during JJA and SON"
in the revised manuscript (lines 254-255).
   We have replaced
"very calm atmosphere"
with
"very laminar atmosphere"
in the revised manuscript (line 362).

Line 378. Don't start a sentence with "And."

**Revision in the manuscript:**
   We have removed
"And"
in the original manuscript (line 378).

**Response letter (RC3)**

Dear Editor and Reviewers:

We greatly appreciate your efforts in the previous version of the manuscript (Title: Antarctic atmospheric Richardson number from radiosoundings measurements and AMPS, Manuscript ID: acp-2022-352). We have made point-to-point responses to all the comments/suggestions raised in your review reports and made the corresponding revisions in the context. All the replies in this document are colored in blue, and the revisions/changes in the revised manuscript are marked in red.

--------Reviewer Comments--------

Recommendation: Reject

General Comments

This paper aims to evaluate the performance of the Antarctic Mesoscale Prediction System (AMPS) model against routine radiosonde observations from three sites across Antarctica. More specifically, the authors evaluate the differences between simulated and observed values of the reciprocal of the Richardson number, Ri, between March 2021 to February 2022. While this is an interesting topic and the methods appear sound, it is the opinion of this reviewer that there are issues with the scope, motivation, and interpretation of the results that would require substantial revisions to make appropriate for publication in Atmospheric Chemistry and Physics.

Fatal Flaw

The primary issue with this study is its lack of analysis or contextualization of results that warrant dissemination in a journal such as Atmospheric Chemistry and Physics. This issue stems from its introduction, which does not adequately frame the motivation for this study apart from generally calculating differences in Richardson number between model output and radiosonde observations. For example, there is little physical insight into the mechanisms behind these differences, how the choice of model physics affect these differences, or statistical significance for any of the computed statistics. The primary conclusions of this study seem only to comprise a report of profiles of 1/Ri differences between an experimental model and radiosondes, a collection of seasonal cross-sections from the model, and a brief comparison of boundary-layer height estimates. This study reads more like a technical report than a journal article with well-defined research questions based on physical mechanisms and addressing a pronounced gap in the literature. Without such analyses, it is difficult to recommend publishing in Atmospheric Chemistry and Physics, whose scope is "…focused on studies with important implications for our understanding of the state and behaviour of the atmosphere". Therefore, it is this reviewer's opinion that this manuscript requires too significant of changes throughout to 1) align with the scope of ACP and 2) provide substantial contributions to the existing literature such that the paper should be rejected with re-submission to another journal encouraged.

**Response:**

We appreciate the reviewer's comments and have revised our manuscript accordingly. We have added analysis of results and aligned them with the scope of ACP. In addition, we have also discussed the physical mechanisms to provide substantial contributions. The main revisions include:

(1) Investigating how the discrepancies between the models and measurements depend on the atmospheric conditions (e.g., the model performance under different potential temperature gradients and wind shear);

(2) Relating the Richardson number distribution with some large-scale phenomena or local-scale dynamics (katabatic winds, polar vortices, convection, gravity wave, etc.) in Antarctica, and investigating the underlying physical processes of triggering atmospheric turbulence.

We present the detailed revisions by responding to your comments below.

Major Comments

1. This manuscript would greatly benefit from a dedicated English language review to ensure the authors' points are properly worded and explained. There are numerous instances throughout the paper where rewording/restructuring sentences and paragraphs would highly improve the readability. Some (but not all) of these instances include lines 46—57, 195—210, 281—284, 302—304, and 383—384.

**Response:**

Thanks for the Reviewer's comments. We meticulously examined each sentence in the paper and enhanced the clarity and precision of the English expression to the best of our ability.

**Revision in the manuscript:**

We have replaced

"And so far, monitoring a wide range of atmospheric turbulence over the Antarctic continent is still tremendous difficult at present, while the atmospheric Richardson number ( $Ri$ ; a critical parameter judging the possibility of the turbulence could be triggered) is easier to obtain, as it can be calculated by the routine meteorological parameters (temperature and wind speed). However, a precise evaluation of atmospheric models to forecast atmospheric $Ri$ in Antarctica is sketchy. Nevertheless, the analyses from the European Centre for Medium-Range Weather Forecasts (ECMWF) have been used to calculate atmospheric $1/Ri$ (the inverse of the Richardson number) by Geissler and Masciadri (2006) and Hagelin et al. (2008). However, their researches have some specific shortages (or problems that need further study): (1) A direct comparison of $1/Ri$ between the model estimations and measurements has not been conducted. (2) Evaluations of forecast ability have not been done yet, while the forecast function is of great significance for practical application. (3) Variability of $Ri$ at high horizontal spatial resolution has not been given, as previous studies focus on atmospheric profiles at separate sites. (4) A reasonable reference standard for judging the atmospheric stability by the model estimations was not given. To fill these gaps, the scientific goals of this paper are thus as follows."

with

"Presently, monitoring a wide range of atmospheric turbulence over the Antarctic continent is tremendously difficult, but atmospheric $Ri$ is easier to obtain, as it can be calculated from the routine meteorological parameters (potential temperature and wind speed). However, few studies have evaluated atmospheric models to forecast $Ri$ in Antarctica, because of limited meteorological experiments here. Nevertheless, Geissler and Masciadri (2006) and Hagelin et al. (2008) used the European Centre for Medium-Range Weather Forecasts (ECMWF) analyses to calculate atmospheric $Ri$ in Antarctica. The ECMWF analyses were generated from the data assimilation using observations (P. Lönnberg, 1992), and can provide initial states for numerical models (such as Polar WRF). However, their research has some specific shortcomings (or problems that need further study): (1) They did not compare $Ri$ estimations from forecasts and measurements, while the forecast function is of great significance for practical application (e.g., astronomical observations, aviation safety, optical communication, etc). (2) How model errors of $Ri$ depends on atmospheric conditions has not been analyzed. (3) The correlations between turbulence conditions (indicated by $Ri$ ) and some large-scale phenomena or local-scale dynamics in Antarctica were not fully investigated. (4) A reference standard for judging the probability of triggering turbulence using the model-estimated $Ri$ was not given. To fill these gaps, the scientific goals of this paper are thus as follows:"

in the revised manuscript (lines 35-47).

We have replaced

[revised manuscript text omitted]

We have replaced
"It shows that the atmosphere exhibited strong daily variability. Thus, forecasting $1/Ri$ is necessary if one wishes to avoid a turbulent atmosphere in Antarctica, instead of counting only on the displayed statistical results. Moreover, unstable atmospheres are likely to be triggered over the ocean, moving toward the Antarctic Plateau and becoming stable. This was likely due to the obstruction of the high plateau, and the atmosphere above it behaved calmly."
with
"It shows that the atmospheric conditions are variable, and a significant transition between laminar flow and turbulent flow could occur at any time. Some activities in Antarctica require a non-turbulent atmosphere, such as astronomical observations (Burton, 2010) and aviation safety (Gultepe and Feltz, 2019). Therefore, real-time forecasting of the Richardson number is important and helpful, rather than relying solely on the statistical results presented in this study. Furthermore, the video shows that atmospheric turbulence is likely to be triggered over the ocean, moving toward the Antarctic Plateau and weakening. This may be due to the obstruction of the high plateau, which creates a calm atmosphere above it."
in the revised manuscript (lines 300-305).

We have replaced
"One can observe that the AMPS-forecasted $PBLH$ is thin over the plateau, especially for the summits (e.g., Dome A and Dome C), which agrees with previous studies (Swain and Gallée, 2006). While thick $PBLH$ are shown near the escarpment region, as seen by the dump in ~67°S (up) in Fig. 10c and ~68°S (down) in Fig. 11c. This corresponds to the thick depth of the large $1/Ri$ shown in enlarged drawings (Figs. 8 and 9)."
with
"A thin $PBLH$ over the plateau can be observed, especially at the Domes (e.g., DA and DC), which is consistent with previous studies (Swain and Gallée, 2006). In contrast, a thick $PBLH$ is shown near the escarpment region (e.g., ~68°S, 122.5°E in Fig. 10c; this corresponds to the relatively low $Ri$ near the ground in A11 area in Fig. 8)."
in the revised manuscript (lines 322-324).

We have replaced
"The overall results show that the AMPS can forecast the behaviour of $Ri$ and could be applied to practical operations such as astronomy in Antarctica, which is interested in the impacts of atmospheric turbulence (Burton, 2010)."
with
"The overall results show that the AMPS can forecast a realistic behaviour of $Ri$, and the turbulence conditions in Antarctica are well revealed; furthermore, some practical operations that want to avoid a

turbulent atmosphere — such as astronomical observations (Burton 2010), aviation safety (Gultepe and Feltz 2019) and free space optical communication (Jianjun Yin et al. 2017) — can apply the AMPS-forecasted $Ri$ .”
in the revised manuscript (lines 367-370).

2. The motivation behind the use of 1/Ri instead of just Ri is not entirely clear beyond citing that another reference did so. One of the citations mentions that 1/Ri provides an easier dynamic range to plot, but in my experience this can be overcome by plotting Ri on a logarithmic axis to maintain the physical interpretation. Please consider changing to use Ri throughout or at least substantiate the use of 1/Ri more thoroughly.

**Response:**

Thanks for the Reviewer's suggestions. We have changed to use $\log_{10}(Ri)$ in the entire manuscript, including the text and all figures (see Figs. 4 to 6 and 8 to 11 in the revised manuscript).

3. In general, there is a lack of substantial analysis or interpretation of the results presented that contributes to addressing gaps in the literature. I would like to see the following addressed to improve upon this aspect of the manuscript.
a. What are some potential underlying reasons for the discrepancies between the models and obseravtions? For example, how does the choice of model physics impact the resulting dynamic stability in simulated profiles? How does the physics in the AMPS model differ from the ECMWF analyses used by Hagelin et al. (2008)? How well does the model perform against other observations in general? How do errors depend on model forecast lead time?

**Response:**

Thanks for the Reviewer's comments.
1. There are many potential underlying reasons that affect the model performance, including terrain environment (analyzed by three different sites: MM, SP, and DC), atmospheric changes (analyzed by four seasons), and model configurations (model physics, grid resolution, etc.). However, the model physics was not well analyzed in our original manuscript (as the sensitivity test for model physics seems to be beyond our study in this paper). Nevertheless, we would like to give a discussion regarding this topic in the revised manuscript. We find that the AMPS-forecasted $Ri$ may be improved by replacing its default microphysics scheme (WSM5C) by Morrison scheme, this is based on our statistical results: the $R_{xy}$ of AMPS-forecasted $\log_{10}(Ri)$ would be higher when the $RMSE$ of potential temperature and wind speed are smaller. Moreover, Hines et al., (2019) showed that the simulation using Morrison scheme presented smaller $RMSE$ for temperature and wind. In sum, higher $R_{xy}$ for $\log_{10}(Ri)$ may be achieved using the Morrison scheme, as the Morrison scheme simulated dynamic stability would be less variable ($RMSE$ for temperature and wind is smaller). We have added these comments in the revised manuscript. What's more, we have investigated how model errors depend on the atmospheric conditions corresponding to different potential temperature gradients and wind shear (see Fig. 6 in the revised manuscript).
2. There is a significant difference between the ECMWF analyses and the AMPS outputs, the former is a product of the data assimilation, and the latter is a numerical forecast product. Geissler and Masciadri (2006) pointed out that an "analysis" provided by the ECMWF general circulation model (GCM) is the output of a calculation based on a set of spatiotemporal interpolations of measurements provided by meteorological stations distributed on the surface of the whole world, and by satellites and instruments

carried aboard aircraft. According to my personal view, the physics for ECMWF is static (interpolation), while the model physics for AMPS is dynamic (simulation). The main use of ECMWF analysis is to provide initial conditions for the numerical forecast (P. Lönnberg and D.B. Shaw 1992). In other words, the ECMWF analysis can be used to initialize the numerical model (Polar WRF) in AMPS. We have added these comments in the revised manuscript.

   3. The AMPS performances for some other observations have been examined in previous studies. as listed in Table R1 in this Response letter. Generally, the AMPS is capable of simulating the main trends of these meteorological parameters. These results are not presented in detail in our manuscript, as this is somewhat beyond the scope of our research (our works focus on displaying the $Ri$ results and reflecting turbulence characteristics in Antarctica). Nevertheless, the revised manuscript has cited these works, which include investigating the performances of AMPS in forecasting temperature, wind, precipitable water vapor, cloud, radiation, and heat flux (including Monaghan et al., 2005; Seefeldt et al., 2011; Vázquez B and Grejner-Brzezinska, 2012; Wille et al., 2016; Listowski and Lachlan-Cope, 2017; Hines et al., 2019).

Table R1. AMPS performance for some parameters  (Bias=AMPS-observation)

| | $R_{xy}$ | Bias | RMSE |
|---|---|---|---|
| Precipitation mount (mm) ① | 0.37 | 56 | - |
| Cloud fraction (%) ① | 0.35 | 2 | - |
| 2  m Specific humidity (g kg$^{-1}$) ② | 0.83 | 0.02 | 0.36 |
| Downwelling shortwave radiation ② | 0.92 | 70.4 | 97.3 |
| Downwelling longwave radiation ② | 0.51 | -41.5 | 54.1 |
| Upwelling shortwave radiation ② | 0.94 | 40.3 | 63.9 |
| Upwelling longwave radiation ② | 0.81 | -8.0 | 13.7 |
| Net radiation ② | 0.70 | -3.3 | 18.4 |
| Sensible heat flux ② | 0.76 | 0.7 | 7.7 |
| Latent heat flux ② | 0.82 | -0.8 | 3.5 |
| Heat Flux into the ice ② | 0.38 | -3.2 | 11.5 |

①McMurdo: Monaghan et al., (2005)
②West Antarctic Ice Sheet (WAIS): Hines et al., (2019)

   4. The measurements used in this study were from the radiosoundings, and the data provided by the radiosoundings are usually regarded as a vertical spatial sequence. Therefore, it may not be appropriate to use the radiosonde measurements to evaluate how errors depend on model forecast lead time, while the AWS measurements (time series data) are more suitable for this. The radiosounding was generally launched once a day at the same hour — we have added this comment in the revised manuscript. As for the AMPS, only the 12-33-h (i.e., a day) forecasts from each of the AMPS forecasts are joined to make an annual time series. Hence, it is hard to evaluate the model performance for forecast lead time, since there is generally only one profile available for comparison from each of the AMPS forecasts.

**Revision in the manuscript:**
   We have added

"Table 2 also shows an interesting result: the $R_{xy}$ of $\log_{10}(Ri)$ is higher when the $RMSE$ of $\theta$ and $V$ are smaller. Moreover, Hines et al. (2019) showed that using the Morrison microphysics scheme in the numerical model resulted in a smaller $RMSE$ for temperature and wind, than the default scheme (WSM5C) in AMPS. Therefore, we may conclude that replacing WSM5C with Morrison could improve

the AMPS-forecasted $\log_{10}(Ri)$. In other words, using Morrison may lead to higher $R_{xy}$ for $\log_{10}(Ri)$, as it simulates dynamic stability with less variability (the *RMSE* for temperature and wind could be smaller). On the other hand, larger *RMSE* for $\theta$ and $V$ are mainly found during cold months (JJA, SON), indicating that winter dynamic stability is more variable (similar to Bromwich et al., 2013).

Table 2 summarises that the $\log_{10}(Ri)$ was overestimated by the AMPS at each site for every season (all *Bias* are positive). This may be due to some local-scale dynamics not being represented properly (see Fig. 6, which will be discussed later). From another perspective, the model results were generally smoother than the measurements, and the atmosphere is less favorable for the occurrence of turbulence under slowly changing meteorological parameters, then the AMPS-forecasted $Ri$ could be larger.

Table 2. Statistical evaluations of the potential temperature ($\theta$), wind speed ($V$), and logarithmic Richardson number ($\log_{10}(Ri)$) forecasted by the AMPS when compared with the results from radiosonde measurements.

| Season | McMurdo | | | | South Pole | | | | Dome C | | | |
|---|---|---|---|---|---|---|---|---|---|---|---|---|
| | MAM | JJA | SON | DJF | MAM | JJA | SON | DJF | MAM | JJA | SON | DJF |
| $\theta$ : $R_{xy}$ | 0.99 | 0.99 | 0.99 | 0.99 | 0.99 | 0.99 | 0.99 | 0.99 | 0.99 | 0.99 | 0.99 | 0.99 |
| $\theta$ : *Bias* | -0.32 | -0.61 | -0.10 | -0.23 | -1.30 | -1.41 | -1.45 | -0.94 | -0.74 | -0.58 | -0.25 | 0.19 |
| $\theta$ : *RMSE* | 1.82 | 1.91 | 1.78 | 1.56 | 2.65 | 2.88 | 3.48 | 2.43 | 4.28 | 2.22 | 2.38 | 1.76 |
| $V$ : $R_{xy}$ | 0.85 | 0.89 | 0.95 | 0.90 | 0.86 | 0.84 | 0.89 | 0.90 | 0.92 | 0.92 | 0.97 | 0.95 |
| $V$ : *Bias* | -0.25 | -0.25 | -0.59 | -0.67 | -0.16 | -0.29 | -0.64 | -0.40 | -0.62 | -0.43 | -0.24 | -0.50 |
| $V$ : *RMSE* | 3.16 | 3.63 | 3.23 | 2.81 | 2.52 | 2.90 | 2.69 | 2.37 | 2.50 | 2.94 | 2.69 | 1.89 |
| $\log_{10}(Ri)$ : $R_{xy}$ | 0.75 | 0.65 | 0.68 | 0.78 | 0.61 | 0.50 | 0.56 | 0.70 | 0.51 | 0.45 | 0.50 | 0.66 |
| $\log_{10}(Ri)$ : *Bias* | 0.41 | 0.32 | 0.33 | 0.47 | 0.36 | 0.23 | 0.29 | 0.35 | 0.45 | 0.39 | 0.30 | 0.46 |
| $\log_{10}(Ri)$ : *RMSE* | 0.90 | 0.86 | 0.88 | 0.93 | 0.90 | 0.84 | 0.85 | 0.90 | 0.93 | 0.86 | 0.81 | 0.91 |

"

in the revised manuscript (lines 198-209).

We have added
"The ECMWF analyses were generated from the data assimilation using observations (P. Lönnberg, 1992), and can provide initial states for numerical models (such as Polar WRF)"
in the revised manuscript (lines 39-41).

We have added
"The performances of AMPS in forecasting temperature, wind, precipitable water vapor, cloud, radiation, and heat flux have been examined in previous studies (Monaghan et al., 2005; Seefeldt et al., 2011; Vázquez B and Grejner-Brzezinska, 2012; Wille et al., 2016; Listowski and Lachlan-Cope, 2017; Hines et al., 2019)."
in the revised manuscript (lines 25-27).

We have added
"Generally, the radiosonde was launched once a day at the same hour (sometimes twice a day at MM and SP). In total, 518, 508, and 340 profiles are available at MM, SP, and DC from 2021 March to 2022 February."
in the revised manuscript (lines 83-85).


b. If the model biases can be related to parameterization schemes, what are the implications for global numerical weather prediction models? Are there underlying stable atmospheric boundary layer or marine boundary layer processes that need be more accurately accounted for in global/regional models?

**Response:**

Thanks for the Reviewer's comments. If the model biases can be related to parameterization schemes; this statement, in our opinion, implies that the parameterization schemes would perform similarly under similar atmospheric conditions. In other words, since the parameterization schemes of AMPS are fixed, the model biases could be mainly determined by the atmospheric conditions. Therefore, we present the performance of AMPS under different atmospheric conditions (see the new Figure 6 in the revised manuscript, the old Figure 6 was deprecated); Fig. 6 shows the model biases for different potential

temperature gradients and wind shears. For instance, one can see that $\log_{10}(Ri)$ is overestimated in relatively unstable atmospheres and underestimated in very stable conditions (Fig. 6a). Regarding strong wind shear conditions, AMPS would underestimate the strength of the wind shear (Fig. 6b). If the model aims for a more accurate forecast of $\log_{10}(Ri)$, the biases under these atmospheric conditions need to be corrected. We have added comments on these results in the revised manuscript.

**Revision in the manuscript:**

We have added

"

[Figure]

Figure 6. Performance of the AMPS under different atmospheric conditions. (a) and (b) are respectively the case of potential temperature gradient ($G=\partial\theta/\partial z$) and wind shear ($S=\left[\left(\partial u/\partial z\right)^2+\left(\partial v/\partial z\right)^2\right]^{1/2}$). $G$ (color of the bin in (a)), $S$ (color of the bin in (b)), $\Delta G$ (stem above the bin in (a)) and $\Delta S$ (stem above the bin in (b)) are presented using the median value for each 0.5×0.5 bin of $\log_{10}(Ri)$.

Fig. 6 shows the AMPS performance under different potential temperature gradient ($G=\partial\theta/\partial z$) and wind shear ($S=\left[\left(\partial u/\partial z\right)^2+\left(\partial v/\partial z\right)^2\right]^{1/2}$). The statistical results presented in Fig. 6 were counted based on all the collected data points at the three sites (MM, SP, and DC) for an entire year. One can see that the $Ri$ was overestimated by the AMPS at unstable atmosphere (see light blue bin in Fig. 6a), where the AMPS has overestimated the potential temperature gradient (i.e. $\Delta G > 0$). But for strong temperature inversion (see dark red bin in Fig. 6a), the AMPS has underestimated the $G$ and $Ri$. As for strong wind shear conditions (see dark red bin in Fig. 6b), when the $Ri$ is small (basically corresponding to a near-surface layer with a high probability of triggering strong turbulence, as in Fig. 4), the AMPS has underestimated the intensity of wind shear ($\Delta S < 0$). This may be caused by the AMPS has underestimated the wind speed near the ground (as in Fig. 3). In sum, if the model aims for a more accurate forecast of $Ri$, the biases under these atmospheric conditions need to be corrected."
in the revised manuscript (lines 217-225).

c. The monthly bulk statistics provided are informative, but could be expanded upon for emphasis. Please provide case studies from, e.g., when or where the model performed particularly well or poorly to determine if there are other large-scale phenomena biasing the model or if local-scale dynamics are not being represented properly.

**Response:**

Thanks for the Reviewer's suggestion. Providing case studies is a good idea. We have investigated the atmospheric conditions when $R_{xy}$ of $\log_{10}(Ri)$ was maximum and minimum. Table 2 (in the revised

manuscript) shows that $R_{xy}$ of $\log_{10}(Ri)$ is the largest (0.77) at MM for DJF and smallest (0.45) at DC for JJA. We discovered a noteworthy phenomenon when analyzing these two cases: the atmosphere at MM during DJF (polar summer) is the most unstable case (median $[\theta_{1000\,m}\text{-}\theta_{0\,m}]/[1000\,m-0\,m]$ equals to 0.0038), while the atmosphere at DC for JJA (polar winter) is the most stable condition (median $[\theta_{1000\,m}\text{-}\theta_{0\,m}]/[1000\,m-0\,m]$ equals to 0.0721). It seems that the AMPS can better capture the trend of $\log_{10}(Ri)$ for unstable atmospheres ($R_{xy}$ is the largest, 0.77). But its *Bias* also increases in this case (Bias is the largest, 0.47); this is because AMPS overestimates $\partial\theta/\partial z$ under unstable atmospheres (Fig. 6a). As for stable atmospheres, the relatively low $R_{xy}$ for $\log_{10}(Ri)$ seems to be consistent with the fact that model errors increase with increasing stability (Nigro et al., 2017).

**Revision in the manuscript:**

   We have added

"The highest $R_{xy}$ is at MM for DJF (0.77) and the lowest is at DC for JJA (0.45). We found that these two cases correspond to the most unstable and stable atmospheric conditions, their median $[\theta_{1000\,m}\text{-}\theta_{0\,m}]/[1000\,m-0\,m]$ equal to 0.0038 and 0.0721, respectively. This suggests that the AMPS can better capture the trend of $\log_{10}(Ri)$ at a relatively unstable atmosphere. However, the *Bias* is the largest (0.47) in the most unstable case. This is because the AMPS overestimated the potential temperature gradient under an unstable atmosphere (Fig. 6a, which will be discussed later). For the stable atmosphere, the lowest $R_{xy}$ for $\log_{10}(Ri)$ seems to be consistent with the fact that model errors increase with increasing stability (Nigro et al., 2017)."

in the revised manuscript (lines 191-197).


**Response:**

We chose the form of equation (2) by considering that the AMPS-forecasted Ri could be improved by reducing its system bias (with $C_1 + C_2 \cdot \log_{10}\left(1/Ri_{AMPS}\right)$) and calibrated by the height factor (with $C_3 \cdot H$). To our knowledge, a calibration on the model-forecasted $Ri$ has seldom been researched. Therefore, it is hard to compare with other results in the literature. However, in the original manuscript, the results from the linear regression model were compared with the original $Ri$ without using it.

In response to your suggestion that our linear regression model reads more like a calibration lab report, we have deprecated it in the revised manuscript. Consequently, we did not further investigate how this

regression depends on seasonality or ambient flow speed and direction. Nevertheless, to provide more scientific analysis in this paper, we have added a new figure (Figure 6) in the revised manuscript, which shows how the bias of the AMPS-forecasted $\log_{10}(Ri)$ depends on the potential temperature gradient and wind shear. This may capture the reader's interest, as we present how model errors depend on atmospheric conditions. If the model aims for a more accurate forecast of $\log_{10}(Ri)$, Figure 6 could offer important guidance for improvement direction. We present our analysis of Figure 6 in the response to Major comment 3(b).

5. The vertical cross-sections in section 4.2.3 are interesting, but I am struggling to understand the value they add to the discussion on model performance versus observations. Observations do not seem to be discussed much at all in this section. Was this supposed to demonstrate the capability of the model to estimate the Richardson number close to the surface? Please tie these results back to the paper objectives of evaluating model performance.

**Response:**

Thanks for the Reviewer's comments.

In Section 4.2.3, we have added a discussion on the model performance versus some large-scale phenomena or local-scale dynamics that may occur in Antarctica: the shear-induced turbulence (katabatic winds, polar vortices), convection (cloud cooling, boundary layer convection), temperature inversion, and the wave-induced turbulence (orographic gravity waves, trapped lee waves, inertia-gravity waves). The possible functional areas of these atmospheric activities are listed in Table 3 in the revised manuscript. These atmospheric activities help us to qualitatively evaluate the AMPS outputs. Overall, the $\log_{10}(Ri)$ patterns are reasonable when related to these atmospheric activities.

The purpose of this revised section is to show when and where turbulence is likely to be triggered in Antarctica with a large temporal and spatial span based on the AMPS outputs (which cannot be done using radiosonde measurements). This section is also dedicated to investigating the underlying physical processes of triggering Antarctic atmospheric turbulence on a large space-time scale.

**Revision in the manuscript:**

We have revised almost all the content of Section 4.2.3 in the revised manuscript (too many changes are made and thus are not shown in this Response letter).

6. Figures 2—5 are interesting, but perhaps present an information overload. I suggest condensing the figures into seasonal median plots instead of monthly medians. Please also consider utilizing a color palette that is more colorblind friendly for all figures in this manuscript (e.g., https://doi.org/10.1175/BAMS-D-13-00155.1).


**Response:**

Thanks for the Reviewer's comments. We have revised the two phrases (as Referee #2 suggested)

**Revision in the manuscript:**

We have replaced

"And so far"

With

"Presently"
in the revised manuscript (lines 35).

We have replaced
"is still tremendous difficult"
With
"is tremendously difficulty"
in the revised manuscript (lines 35).

7. Line 49: The use of the word "sketchy" is too colloquial for discussion in a journal
article. Please rephrase to discuss the relative lack of performance evaluations of models
in Antarctica.

**Revision in the manuscript:**
We have replaced
"However, a precise evaluation of atmospheric models to forecast atmospheric $Ri$ in Antarctica is sketchy."
With
"However, few studies have evaluated atmospheric models to forecast $Ri$ in Antarctica"
in the revised manuscript (line 37).

8. Line 54: Please elaborate on the "practical applications" mentioned at the end of point 2 to improve
the motivation of this study.

**Revision in the manuscript:**
We have added
"(e.g., astronomical observations, aviation safety, optical communication, etc)"
in the revised manuscript (line 43).

9. Line 66: Please remove "or instability" to be more concise.

**Response:**
Referee #2 indicates it should not claim that the presence of turbulence equals an unstable layer.
Instead, clarify that $Ri$ gives insight into the stability of a layer. Then, we would like to say it may give
insight into the atmospheric turbulence in Antarctica.

**Revision in the manuscript:**
We have replaced
"allowing us to evaluate the ability of the AMPS to forecast atmospheric stability or instability."
With
"allowing us to evaluate the reliability of AMPS-forecasted $Ri$ in giving insight into the atmospheric
turbulence in Antarctica."
in the revised manuscript (lines 55-56).

10. Line 91: In the sentence "The balloon scans the atmosphere," please consider replacing
"scans" with "observes" or "measures" for clarity.

**Revision in the manuscript:**
We have replaced
"scans"

with

"meansure"

in the revised manuscript (line 95).

11. Lines 98—109: Please make it more apparent which model forecast times are selected for analysis. Please also add discussion on the choice to use forecasted fields instead of model analysis instances.

**Revision in the manuscript:**

We have replaced

"This study used the AMPS outputs (in original WRF format) for forecast hours 12-21 at three h intervals from the daily AMPS forecasts that began at 12:00 UTC. Thus, our AMPS fields had a spin-up time of 12 h (similar to Hines and Bromwich, 2008; Hines et al., 2019). Finally, the AMPS forecasts for the same period (from March 2021 to February 2022), as the used radiosounding measurements, were downloaded for analysis."

with

"This study used the AMPS outputs (in original WRF format) from the daily AMPS forecasts that began at 12:00 UTC. Parish and Waight (1987) showed large adjustments to the boundary layer fields above an ice sheet before the numerical model began to stabilize after about 10 h. Then, some studies (Hines and Bromwich, 2008; Hines et al., 2019) have discarded the first 12 h forecasts (so-called 12 h spin-up time). Thus, in this study, only the 12-33-h forecasts from each of the AMPS simulations are combined into a year-long (2011 March to 2022 February) output field at 3-h intervals."

in the revised manuscript (lines 106-111).


**Revision in the manuscript:**

We have replcaced

"where"

with

"Where"

in the revised manuscript (line 115).

We have added

"As for wind shear term,"

in the revised manuscript (line 116).

13. Lines 141—143: This information better belongs in the caption for Figures 2 and/or 3.

Please consider moving.

**Response:**

   Thanks for the Reviewer's comments. We have condensed the figures into seasonal median plots as you suggested, so the month information should be removed.

**Revision in the manuscript:**

   We have removed

"The panels in the same row of Figs. 2 and 3 correspond to the same season; the first row: Aut. (Autumn: March, April, and May), second row: Win. (Winter: June, July, and August), third row: Spr. (Spring: September, October, and November), fourth row: Sum. (Summer: December, January, and February)."
in the original manuscript (lines 141-143).

   We have replcaced

"Figure 2. The monthly median for temperature forecasted by the AMPS (solid lines) and temperature difference calculated by the AMPS forecasts minus the radiosoundings measurements, i.e. $\Delta T = T_{AMPS} - T_{Mea.}$ (filled areas)."
with

"Figure 2. The seasonal median of potential temperature ($\theta$) estimated by the radiosonde measurements (solid lines) and potential temperature difference ($\Delta\theta$) calculated by the AMPS forecasts minus the radiosonde measurements, i.e. $\Delta\theta = \theta_{AMPS} - \theta_{Mea.}$ (filled areas). Fall: March-May (MAM); winter: June-August (JJA); spring: September-November (SON); summer: December-February (DJF)."
in the revised manuscript (see the caption for Figure 2, line 154, page 7).

14. Line 154: What months are being referred to when discussing the model and observation differences here?

**Response:**

   Thanks for the Reviewer's comments. This is the temperature difference between different sites (not the difference between the model and observation). This sentence has caused misunderstanding, and we have condensed the figures into seasonal median plots as you suggested, so this discussion is no longer given in the revised manuscript. Nevertheless, you can find the differences between the model and observation for different seasons in Table 2 (in the revised manuscript).

**Revision in the manuscript:**

   We have removed

"Fig. 2 shows that the forecasted temperature profiles of SP and DC in the first 5 km are similar, 15-20 K colder than the MM."
in the original manuscript (lines 154-155).

15. Line 175: Specifically, what heights are these parameters being interpolated to? Maybe this could be depicted in a figure.

**Response:**

   Thanks for the Reviewer's comments. We have depicted the interpolated heights, as shown by the markers in Figures 2-4.

16. Lines 179—182: This paragraph discussing the use of NCL is not necessary for the presentation of results. Please instead discuss how derivatives are calculated (i.e., centered finite differencing, fitting an analytical function, etc.) and how this choice may impact the resulting profiles of 1/Ri.

**Response:**

Thanks for the Reviewer's comments. We have discussed how derivatives are calculated and fit an analytical function for $\log_{10}(Ri)$ profile.

**Revision in the manuscript:**

We have replcaced

"The NCL (NCAR Command Language) can compute Ri from the original WRF outputs, using an NCL function called "rigrad_bruntv_atm"(http://www.ncl.ucar.edu/Document/Functions/Contributed/rigrad _bruntv_atm.shtml). The NCL-computed (not shown) agrees very well with the results obtained using interpolation (i.e. interpolating the hybrid vertical coordinate to the same height series for each site)."
with

"Where $\partial\theta/\partial z$ and $(\partial u/\partial z)^2 + (\partial v/\partial z)^2$ for calculating $Ri$ (see Eq. (1)) were both computed using a centered finite difference operation, as we found that centered difference performed better than forward difference and backward difference (not shown), i.e., better consistency of $Ri$ between radiosoundings and AMPS forecasts can be achieved using centered difference."
in the revised manuscript (lines 164-167).

We have added
"

[Figure]

Figure 5. The polynomial curve fitting of near-ground median profiles of $\log_{10}(Ri)$ from Fig. 4; (a) the $\log_{10}(Ri)$ was estimated by the radiosonde measurements, (b) the $\log_{10}(Ri)$ was estimated by the AMPS forecasts.

The near-ground atmosphere in Antarctica is an important turbulence source, then an analytical function for $\log_{10}(Ri)$ profiles near the ground was fitted to better contextualize the results (as shown in Fig. 5). Fig. 5 shows that the near-ground $\log_{10}(Ri)$ profiles are the "C-C-C" shape. The "concave" structure in the "C-C-C" shape could be attributed by the near-ground jet stream (Mihalikova et al., 2012). A cubic

polynomial function was used (see the upper part of plots in Fig. 5) instead of a logarithmic function, because the "C-C-C" shape seems hard for logarithmic function fitting. Moreover, each fitted curve used all four-seasons data points in Fig. 4, as the seasonal variation are not too significant. Nevertheless, one can see more details about the temporal variation of $\log_{10}(Ri)$ near the ground in Sect. 4.2.3."

in the revised manuscript (lines 210-216).

17. Line 184—185: The sentence beginning "This is because the value of 1/Ri can oscillate…" is vague and potentially misleading. Please clarify what is meant here.

**Revision in the manuscript:**

We have replaced

"This is because, the value of $1/Ri$ can oscillate massively in the atmosphere, and a precise quantification seems less plausible"

with

"This is because, the $Ri$ value can vary massively (by two or more orders of magnitude) in the atmosphere, and a precise quantification seems less plausible."

in the revised manuscript (lines 169-170).

18. Line 194: Please add a reference to a figure or citation to contextualize the discussion at the end of this paragraph.

**Response:**

Thanks for the Reviewer's comments. We have condensed monthly plots into seasonal median plots in new Figure 4 and some new results are presented. We also have given a discussion for the new figure 4.

**Revision in the manuscript:**

We have replcaced

"On the other hand, in a more detailed comparison, for example, in March 2021, both forecasts and measurements show that $1/Ri$ profiles are inclined to the y-axis at ~10 km AGL, and in December 2021, the AMPS forecasts are able to capture a small bump of $1/Ri$ that occurred at around 7 km AGL."

with

"In Fig. 4, one can see that the AMPS-forecasted $Ri$ can identify that the atmosphere above MM tends to be more turbulent ($Ri$ is smaller) than SP and DC. In the vertical height direction, the AMPS forecasts can roughly capture the height that can easily trigger turbulence. For example, one can observe that the $Ri$ from radiosoundings and AMPS forecasts both show small values very close to the ground at DC and the SP, which is per the fact that strong atmospheric turbulence is concentrated within the surface layer above the high plateau (Marks et al., 1999; Agabi et al., 2006). A very calm atmosphere ($Ri$ is large) at high altitudes is also consistent with the results given by Travouillon et al. (2003), Aristidi et al. (2005), Trinquet et al. (2008) and Vernin et al. (2009). On the other hand, the AMPS can reconstruct the near-ground "convex-concave–convex" (hereafter "C-C-C") shaped $\log_{10}(Ri)$ profiles indicated by the radiosonde measurements (see more details in Fig. 5). In terms of time, the AMPS can forecast that the free-atmosphere $Ri$ decreased during spring (SON), this decrease is obvious for MM and DC (where the wind speed are significantly stronger during SON, as in Fig. 3)."

in the revised manuscript (lines 173-183).

19. Line 195: The phrase "…are afraid to conduct quantitative analysis…" is too colloquial and does not accurately portray the gaps in the literature. Please rephrase.

**Revision in the manuscript:**

We have replaced

"Some researchers are afraid to conduct quantitative analysis for the model-estimated $1/Ri$ as it always varies dramatically"

with

"Quantitative analysis for the estimated $Ri$ from the numerical models was generally missed as it always varies dramatically (e.g., Hagelin et al., 2008 focused on the qualitative analysis)."

in the revised manuscript (lines 184-185).

20. Line 210: Please provide additional discussion for why correlations are lower in the winter other than it is harder to collect observations.

**Response:**

Thanks for the Reviewer's comments. As addressed in Major Comments 3(c), the lowest $R_{xy}$ for $\log_{10}(Ri)$ seems to be consistent with the fact that model errors increase with increasing stability (Nigro et al., 2017), where the atmosphere is generally more stable during winter.

**Revision in the manuscript:**
    We have replaced
"through the South Pole and Dome C (shown in Fig. 7a)."
with
"through the South Pole (black circle) and Dome C (black triangle), as shown by the red line in Fig. 7a."
in the revised manuscript (see the caption for Figure 8, line 247, page 13).

    We have replaced
"the red line through Dome A and McMurdo (shown in Fig. 7b)."
with
"the vertical cross-section through Dome A (black star) and McMurdo (black cross), as shown by the red line in Fig. 7b."
in the revised manuscript (see the caption for Figure 9, line 247, page 14).

25. Line 281—283: Please reword this sentence to better portray the importance of forecasting 1/Ri accurately.

**Revision in the manuscript:**
    We have replaced
"shows that the atmosphere exhibited strong daily variability. Thus, forecasting 1/Ri is necessary if one wishes to avoid a turbulent atmosphere in Antarctica, instead of counting only on the displayed statistical results."
with
"It shows that the atmospheric conditions are variable, and a significant transition between laminar flow and turbulent flow could occur at any time. Some activities in Antarctica require a non-turbulent atmosphere, such as astronomical observations (Burton, 2010) and aviation safety (Gultepe and Feltz, 2019). Therefore, real-time forecasting of the Richardson number is important and helpful, rather than relying solely on the statistical results presented in this study. Furthermore, the video shows that atmospheric turbulence is likely to be triggered over the ocean, moving toward the Antarctic Plateau and weakening. This may be due to the obstruction of the high plateau, which creates a calm atmosphere above it."
in the revised manuscript (lines 300-305).

26. Line 297: Please elaborate on the comparisons with sodar observations, this seems to be included without proper contextualization.

**Revision in the manuscript:**

We have replaced

"which is consistent with the SODAR observations"

with

"such variation range is consistent with the SODAR observations"

in the revised manuscript (lines 317-318).

---

## Author Response (AR2)

**Response letter**

Dear Editor and Reviewers:

We greatly appreciate your efforts in the previous version of the manuscript (Title: Antarctic atmospheric Richardson number from radiosoundings measurements and AMPS, Manuscript ID: acp-2022-352). We have made point-to-point responses to all the comments/suggestions raised in your review reports and made the corresponding revisions in the context. All the replies in this document are colored in blue, and the revisions/changes in the revised manuscript are marked in red.

--------Reviewer Comments--------

General Comments
The author has sufficiently addressed my comments on the prior manuscript draft. I have only a few remaining minor comments.

**Response:**

We appreciate the reviewer's comments and advice. We have revised the manuscript accordingly.

Minor comments

Line 33: I don't understand what this means: "corresponding to the measured astronomical seeing is small"

**Response:**

A small astronomical seeing means the intensity of optical turbulence is small, optical turbulence is the effects of atmospheric turbulence on wave propagation. For example, Atmospheric turbulence is a major problem in optical astronomy as it drastically reduces the angular resolution of telescopes (Roddier, 1981).

**Revision in the manuscript:**

We have replaced

"the simulated $Ri$ basically behaved as expected as the $Ri$ is generally large when the atmosphere is less turbulent (corresponding to the measured astronomical seeing is small; Yang et al., 2021)"
with

"the simulated $Ri$ behaved as expected since the $Ri$ is generally large when the disturbance effects of atmospheric turbulence on wave propagation (called optical turbulence) are weak (Yang et al., 2021)."
in the revised manuscript (lines 32-33).

**References:**

Roddier F.: The Effects of Atmospheric Turbulence in Optical Astronomy, Progress in Optics, 19, 281-376, 10.1016/S0079-6638(08)70204-X, 1981.

Line 88: I don't think it is necessary to mention what Vaisala sensor used to be used or how its sensitivities compare to the Vaisala RS41, if measurements from the older version were not utilized in this study, as this makes the message more convoluted.

**Response:**

Thank you for the Reviewer's suggestions. We have removed comments on the older version (Vaisala RS41) to avoid making the message more convoluted.

**Revision in the manuscript:**
    We have deleted
"Vaisala RS41 radiosondes have gradually replaced an older version (Vaisala RS92) starting in late 2013. These two radiosondes agree well with global average temperature differences <0.1-0.2 K in the lower stratosphere, but  RS41 appears to be less sensitive than RS92 to changes in solar elevation angle (Sun et al., 2019). Besides, RS41 (1-1.5\% dry bias) has better performance than RS92 (3-4\% dry bias) relating to the infrared atmospheric sounding interferometer as a practical reference (Sun et al., 2021)."
in the original manuscript (lines 88-92).

Line 93: Not necessary to write out Richardson number and boundary layer height, as these have previously been defined with acronyms.

**Response:**
    This sentence located in Line 93 of the original manuscript has been omitted. Please refer to the response below for further information.
    Nevertheless, we have utilized abbreviations for Richardson number and boundary layer height in other instances.

**Revision in the manuscript:**
    We have replaced
"The Richardson number is used to determine the boundary layer height using a critical value"
with
"The $Ri$ is used to determine the $PBLH$ using a critical value"
in the revised manuscript (line 299).

Line 92-94: What is the point of this sentence? Clarify the significance of the fact that the Ri and PBLH from the radiosondes are positively correlated with that from reanalysis.

**Response:**
    It seems that the significance of this sentence is not obvious, Then we have deleted it.
    Nevertheless, we would like to explain the point of this sentence here. Reanalysis can be used as the model initial and boundary conditions for AMPS (or AMPS inputs). We previously think it may be worth mentioning how it performs when compared with the results from the radiosondes. However, this content does not stick to the topic of this paper, as this paper focuses on evaluating the AMPS outputs, then this description has been removed.

**Revision in the manuscript:**
    We have deleted
"Near-global radiosonde measurements have been used to calculate the Richardson number and derive the boundary layer height, which is positively correlated with the results of four reanalysis products"
in the original manuscript (lines 92-94).

Figure 4: In every other x-axis label, it says logRi(10) – shouldn't this be log10(Ri)?

**Response:**
    We have revised the x-axis label in Figure 4 and replaced $\log_{Ri}(10)$ with $\log_{10}(Ri)$.

Figure 8 caption: not necessary to state "(instead of {theta})"

**Response:**

We have deleted "(instead of {theta})".

**Revision in the manuscript:**

We have deleted

"(instead of $\theta$)"

in the original manuscript (Figure 8 caption).

Line 302: Should use abbreviation for Richardson number instead of writing it out.

**Response:**

We have used an abbreviation for Richardson number.

**Revision in the manuscript:**

We have replaced

"Richardson number"

with

"$Ri$"

in the revised manuscript (294).